# Landscape Conservation Assessment in the Latin American Tropics: Application and Insights from Costa Rica

**Vassiliki Vlami** [1], **Carlos Morera Beita** [2] **and Stamatis Zogaris** [3,*]

1   Department of Environmental Engineering, University of Patras, 26504 Patras, Greece; vasvlami@upatras.gr
2   Geography School, National University, Heredia 40101, Costa Rica; carlos.morera.beita@una.cr
3   Hellenic Centre for Marine Research, 19013 Anavissos, Greece
*   Correspondence: zogaris@hcmr.gr; Tel.: +30-2291-0-76393

**Abstract:** Landscape quality is an important aspect of conservation and sustainable development, yet holistic assessments of landscapes in the Latin American tropics are scarce. Here we employ an onsite survey across Costa Rica using the Landscape Assessment Protocol (LAP), a rapid assessment method, to assess the conservation condition of landscape views. In a survey of 50 landscape view sites in different parts of the country, LAP's 15 metrics (evaluation criteria) were effective in providing an index for landscape quality showing a gradient of degradation in response to various modern anthropogenic pressures. The response of the index over a variety of landscape types correlates well with the Human Footprint anthropogenic pressure assessment, an independent land degradation index. Urban and peri-urban landscape types showed the most degraded conditions relative to flatland, coastal, and upland types on all metrics. Despite certain subjective attributes, the assessment method seems effective in providing a quality condition index that may assist in quality characterization and in promoting participation in landscape interpretation, landscape literacy, and landscape-scale conservation initiatives, especially in a region where landscape views (scenic resources) are threatened by widespread land-use changes. Finally, recommendations are made for the further application and testing of LAP, specifically for use in the neotropics.

**Keywords:** conservation; applied geography; landscape ecology; protected areas; biodiversity; land management; Landscape Assessment Protocol

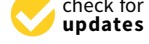



## 1. Introduction

Landscape assessment is a complex undertaking and often produces divisive discourse. Differing perspectives from different disciplines have produced various methods of landscape study [1,2]. Most methods of landscape assessment are from the Global North, with traditions hailing from Europe and North America [3,4] and much development in Australia as well [5,6]. In the Global South, research and educational applications focusing on landscape qualities are scant [7]. Even in the neotropics, where landscape ecology has been an important biodiversity conservation topic, and despite increased academic attention, holistic investigations of landscape quality are scarce [8–11]. Until recent years, the inherent benefits provided by landscapes, such as scenic, aesthetic and other cultural services, were rarely discussed in Latin America, and few methodological tools for their onsite assessment have been introduced [12,13]. In this paper, we explore the issue of landscape quality surveying through an application in the neotropical country of Costa Rica.

Costa Rica has a unique place in Latin American nature conservation history, spearheading active concern and long-term commitment [14]. Costa Rica has been a promoter of many types of protected areas, including pioneering inroads in the selection, delineation, management, and application of early landscape-scale conservation initiatives and biological corridors [15,16]. The importance of the landscape-scale approach has been actively discussed since the mid-1980s [17], and many broad-scale studies were initiated on

biodiversity conservation applications, including efforts for Payment for Environmental Services (PES) schemes [16] and ecosystem services [18]. Conservation science research has provided prescriptions for landscape-scale biodiversity management [19], including guidance in so-called working landscapes [18,20–22]. However, efforts for characterization and assessment of all landscapes, beyond the focus on biodiversity or protected areas, are few and recent, e.g., [23–26]. The issues of scenic landscape resources and holistic landscape quality assessment are still not well developed [12].

Throughout Latin America, both natural and human-modified landscapes have rapidly altered due to recent changes: globalized agriculture, resource extraction, communications networks (including roads), and urban sprawl continue to degrade authentic landscapes, producing novel ecosystems and expanding so-called domesticated landscapes [27,28]. Moreover, various landscapes are modified in different and often subtle ways by humans; some have a much older relationship with traditional land-use patterns; these are sometimes called cultural landscapes. In a global sense, cultural landscapes have long been neglected both within and outside of protected areas [29–31]. Cultural landscapes often hold an important biocultural heritage, an integrated coevolution between biodiversity and traditional human societies [32]. The biocultural heritage usually reflects something "traditional and old", including attributes of biocultural elements in the landscape [33]. The basis for what is considered a cultural landscape of high integrity (or of biocultural heritage) often has historical roots, but the value people assign to it is perceived differently with respect to the different cultures and societal groups [30,33,34]. These are difficult "thorns" in the landscape assessment process and its methodological standardization [3,35]. Subjective philosophical issues blur the consensus on assessment methods [1], and holistic assessment and management in human-modified landscapes also varies among disciplines [36,37].

Without assessment and management, the landscape heritage in Latin America will continue to suffer severe losses to its natural and cultural heritage [38]. This is especially true in the super-biodiverse tropics, such as in Mesoamerica [19,39,40]. Moreover, as should be obvious, landscapes are important for society by providing multiple ecosystem services including scenic and non-material resources critical for economic development, such as tourism, recreation and education. Part of the success of nature conservation in Costa Rica has been an economic incentive towards the promotion of protected areas. This combined biodiversity conservation with a unique nature-branded tourism industry; it has defined the dominant tourism narrative for decades now [41]. However, even the good ecotourism examples may inadvertently promote negative tourism-influenced landscape-scale changes beyond the protected core areas. Examples include difficulties on the fringes of Costa Rica's smaller parks [42], and where small private reserves solely protect "core habitats" but cannot guarantee conservation of the wider landscape [43]. In the last decade, nearly one-quarter of Costa Rica's export income came from the expanding tourism sector, increasingly spreading across a mix of tourist-related development, not just "green" or ecotouristic [44,45]. Many land management difficulties develop outside of protected areas especially. So, landscape conservation beyond the protected areas is important because it protects wider "landscape" cultural values and multi-faceted ecosystem services of high economic importance. These influence both the economy and the quality of life for both locals and visitors. In this way, all landscapes need to be inventoried assessed, monitored, and understood just like other life-giving resources.

In the last two decades, landscape study has seen new developments. For example, new perspectives were pioneered in Europe, especially after the European Landscape Convention (ELC) was enacted in 2000. In recognizing that landscape assessment should no longer focus only on outstanding landscape sites, the ELC initiated a paradigm shift in science-policy applications by promoting the study of all landscapes [46,47]. A widely used landscape description protocol has been the Landscape Character Assessment (LCA) [46,48]. LCA was initially developed to delineate and characterize all landscapes in the UK and has since been applied in various ways throughout nearly all of Europe, often in the name of ELC commitments [37,49]. Outside of Europe, North America, and Australia, assessment

methods are still mostly based on the valuation and designation of specific landscape sites, rarely in efforts towards an assessment or characterization of all landscapes, and landscape views in particular [50].

Generally, there is a scarcity of standard methods using onsite assessment protocols, with most work utilizing remote sensing or location photographs. Few onsite assessment protocols have been developed or widely used; most applications focus on broader geographical indicators [37]. Available protocols focus on the quality of particular landscape formations, such as for river valleys [51], beaches [52], or LCA approaches in particular regions [7]. Recent advances have included ways to measure the cultural ecosystem services provided by cultural landscapes, [53], GIS-assisted assessments [3], and public participation in assessment efforts [54]. The onsite "rapid assessment" protocols could be interesting for scientific monitoring and citizen participation [55,56].

We decided to explore landscape quality in a tropical humid climate area through an onsite approach using the Landscape Assessment Protocol (LAP), a rapid landscape assessment survey method [57]. This method, first published in 2016 [58], has been developed and tested primarily in temperate and Mediterranean climate areas, but has not been formally tested in the tropics. Here, we aimed to apply a country-wide survey of landscape views using the original LAP method, and we critique the application in a wide variety of landscape types in Costa Rica. We aim to see if there are benefits in utilizing such a rapid assessment tool for a baseline inventory of landscape conditions, landscape degradation descriptions, and how this may be of use in landscape conservation in the Latin American tropics.

## 2. Materials and Methods

### 2.1. Assessment Premises and Philosophy

The LAP is a rapid assessment method for surveying the conservation status of landscape views. It has already been employed in university education, summer schools, and environmental assessments in Europe and Asia [34,59], and has recently been used in South and Central America as well [13,60]. The LAP's practical and rather simple holistic approach is described in detail in a paper by Vlami and colleagues [57], but important premises of the application should be reiterated here:

- The approach evaluates landscape view conservation status by examining 15 evaluative criteria (or metrics) for landscape quality. The protocol utilizes different evaluation criteria (metrics) that respond to modern anthropogenic degradation. Each metric refers to different landscape-scale attributes, each having a reference condition state (the "excellent" state, or 10) and a gradient to total degradation conditions (down to "bad", or 0). The assessor rates the quality of landscape views, not landscape areas (i.e., previously cartographically delineated parcels of land). Only what is perceivable from a viewpoint is assessed.
- Each metric is a quality or characteristic element of the landscape that is known to predictably alter when influenced by human-induced pressures or changes, thus reflecting the quality of a different aspect of the "landscape system". The metrics cover six different thematic categories: land use, human structures, pollution, biodiversity, ecosystem integrity, and aesthetic quality.
- Each metric is scored by the assessor (or assessors) onsite using a field card (Figure 1) and scoring criteria guidance sheet (Appendix A, Figure A1). A landscape view site must have at least a 180-degree view of the surrounding landscape (assessors can wander up to a 50 m radius from the viewpoint during the assessment). The assessor scores each metric based on the scoring criteria guidance sheet narrative. This code guides the evaluation through an easy-to-use descending score level (i.e., 10 to 0). If an assessor is uncertain how to assess a metric, it should be left without a score. Finally, the LAP provides an integrated semi-quantitative index summarizing the conservation status of the assessed landscape; the LAP index is expressed as a 5-to-1 (excellent to

bad) characterization of the landscape view. A trained assessor completes the LAP in about 10 min.

## Landscape Assessment Protocol - LAP

Site Name.........................................Coordinates...................................................

Recorder Name........................... Date.........................Time Start...................Time Stop..................

Primary landscape anthropogenic pressures.........................................................................

### *Metrics to be scored (10-0 scale)*

☐ Land Use Pattern (L)          ☐ Landscape Attractiveness (A)

☐ Vegetation (I)                ☐ Smellscape Pleasantness (P)

☐ Flora (B)                     ☐ Wildlife & Wildlife Habitat (B)

☐ Road Network (H)              ☐ Buildings (H)

☐ Modern Anthropogenic Interference (H)

☐ Pollution, Garbage & Debris (P)

☐ Agriculture (L)

☐ Livestock Grazing (I)

☐ Hydrologic Alteration (I)

☐ Shorelines &/or Riparian Conditions (I)

☐ Soundscape Quality (A)

**LAP CI**

| Sum | Overall Score (Total divided by number scored X 10) |
|-----|-----|

### THEMATIC CATEGOREIS (METRICS)

LAND-USE (L)
HUMAN STRUCTURES (H)
POLUTION (P)
BIODIVERSITY (B)
ECOSYSTEM INTEGRITY (I)
AESTHETIC QUALITY (A)

### LAP CI - INDEX RATING

| LAP INDEX | COLOR & SCORE | QUALITY CLASS |
|-----------|---------------|---------------|
| ≥85 | Dark Green  5 | Excellent |
| 70–84 | Green  4 | Good |
| 50–69 | Yellow  3 | Moderate |
| 31–49 | Orange  2 | Poor |
| ≤30 | Red  1 | Bad |

**Figure 1.** The LAP field form's scoring card with the 15 metrics—the first page of the field protocol, where they are scored on a 10–0 scale (**Left**). In the grey inset (**Right**): a summary interpretation reiterates that there are six thematic categories to each metric (shown in parentheses beside each metric). Inset at Right: the original rating scale class boundaries and quality class characterizations of the LAP conservation index (LAP CI), following Vlami and colleagues 2019 [58]).

The approach that the LAP applies promotes a merging of the biocentric and socio-cultural paradigms in assessment traditions, following the tenets of landscape ecology and the study of landscape history and natural history, incorporating techniques from site-based rapid bioassessment surveys. Indices and rapid assessment approaches have been widely developed for ecosystem monitoring using specific evaluation criteria (such as tested metrics) for over four decades now; most are founded on an understanding of ecological integrity [61]. The rationale for utilizing visible and perceptible metrics onsite is widespread in biologically-based rapid assessments of ecosystem conservation conditions [62]. In contrast to specific ecosystem types, landscapes are often highly dynamic, complex, and heterogeneous "systems", and when the human aspect enters the framework, there are challenges in systematizing such assessments [63]. As Daniel (2001) [1] purports: "landscape quality derives from the interaction between biophysical features of the landscape and perceptual/judgmental processes of the human viewer". Through an onsite assessment method, we acknowledge that "quality" is supported by both landscape conditions and the perceptual processes the landscape view "evokes" in the assessor.

### 2.2. Study Area

Costa Rica is renowned for its remarkable landscape diversity. A thin and high cordillera with many volcanos creates a central backbone splitting the small country between the Pacific and Caribbean slopes; the highest peak, Cerro Chirripó, is 3819 m above sea level. Most of the country was originally covered by forest. Humid tropical rainforest

dominated both west and east of the cordillera, while the northwest had seasonally dry lowland forests and woodlands and the high mountains hosted vast broadleaved cloud forest and highland scrublands [64]. This biological cross-roads of Mesoamerica was inhabited by indigenous First Nations before coming under Spanish rule in the 16th century; for centuries, development was limited to the central plateaus. Costa Rica now has a population of about five million in an area of 51,060 km². The country has experienced extensive landscape changes during the last five decades, with industrial monoculture expansion (banana, pineapple, palm oil, etc.) in many parts of the country, while coffee and several other fruit trees and garden vegetables are often in mixed landscapes; many traditional small holdings still exist. Landscape changes also involve urban sprawl and expanding road networks; however, there has been widespread recent regeneration of felled forests [65–67]. Tourism has been a growing industry but is still fairly localized. Efforts for conservation areas and parks began earlier than other tropical Latin American countries [68], with remarkable success (now covering 25% of land and nearly 30% of marine areas). In this survey work, we assessed landscape conditions in no less than six different terrestrial ecoregions (following Olson and Dinnerstein 2002 [69]), in varied landscapes, including coastal, flatland, upland, and urban environments.

### 2.3. Application in Costa Rica: Specific On-Site Methods

The landscape view being assessed by our method is "a portion of a territory that the eye can comprehend in a single view"; this is one of the many definitions of landscape [1]. The human perception of landscapes is determined by the location of the viewpoints, and views should be used which cover the range of landscape types within the wider study area or region [6]. Site selection (landscape viewpoints) is often unavoidably biased and usually impeded by practical constraints and accessibility problems. In this case, site selection was based on the following site attributes, following the original protocol directions [58]: (a) the potential for a wide view, spanning at least 180 degrees; (b) selection based on representativeness of the landscape area, i.e., avoiding the inclusion of repetitive landscape scenes and making an effort to cover completely different vista/view types and conditions; (c) a selection of sites fairly far apart geographically (most sites are at least 1 km apart). Although an effort to select various landscape types was made, finding panoramic landscape views proved challenging in the humid high-forest conditions. The high tree stands often impeded panoramic views. This was also difficult in higher elevations due to weather conditions, as this survey was conducted during the wet season (i.e., fog and rain did not allow adequate views in many highland areas). Despite this shortcoming, it should be said that the LAP is best applied in areas that have at least some cultural/anthropogenic disturbance; most upland wilderness areas would of course score as "excellent" on nearly all metrics.

This survey was executed by two experienced landscape connoisseurs, co-founders of the LAP (V.V. and S.Z.), who selected 50 viewpoint sites in the Pacific, montane, and peri-urban areas, as well as the country's Caribbean slopes. The survey was completed in one continuous 13-day road trip (6.08 to 18.08.2021) across the country. Figure 2 summarizes the survey application.

Assessors may complete LAP assessments independently side-by-side or work on one LAP together [58]; in this application, the latter was applied and a consensus was reached for every single LAP assessment at each site. Only the landscape view was assessed with everything completed onsite; no other resources for assessing landscape view were utilized (i.e., aerial images etc.).

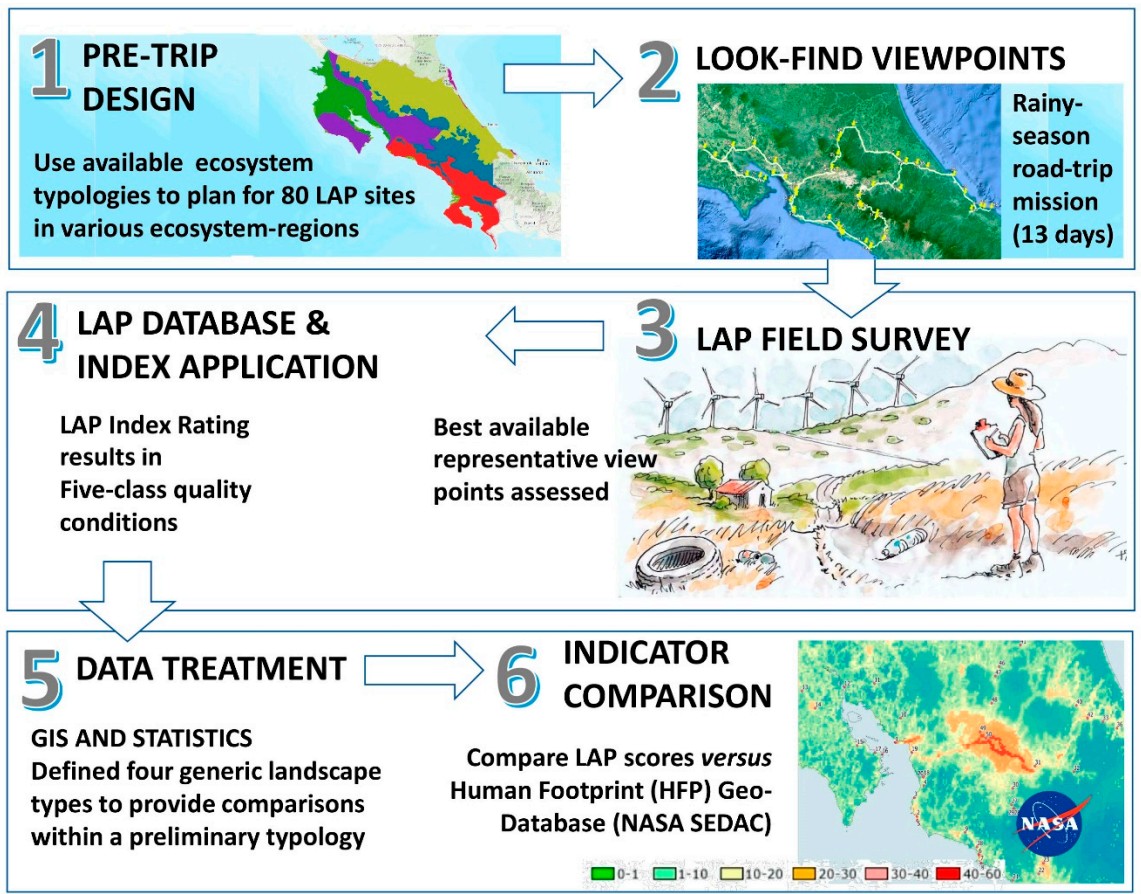

**Figure 2.** Summary of LAP application in Costa Rica: (**1**) designing the survey to cover different ecosystem types throughout the country; (**2**) searching for viewpoint possibilities to apply the LAP; (**3**) assessing selected landscape views onsite; (**4**) database, data entry, and index computation; (**5**) cartographical and statistical analysis; and, (**6**) insights from the results of other indicators. (Terrestrial ecoregion delineation (step 1) based on Olson and Dinnerstein 2001 [69], retrieved from https://databasin.org/ uploaded by http://consbio.org, accessed on 1 February 2022).

### 2.4. Statistical Analyses and Validating Assessments

Data management included quality control and assurance checks upon data entry, and resulted in a simple MS excel matrix (Appendix A, Table A2). Descriptive statistics where applied using SPSS 18.0. Multivariate statistical analysis was carried out with Primer 6β. Open-source GIS was utilized for geographical analyses and cartography [70].

Although validation was not a key aspect of this study, efforts were made after the study to locate environmental degradation databases to compare with the LAP assessments. The Human Footprint (HFP), a human pressure map from NASA's Socioeconomic Data and Applications Center (SEDAC), provided a global map of the cumulative human pressure on the environment at a spatial resolution of ~1 km$^2$ [71]. This cartographic geodatabase measures human pressure using eight variables, including built-up environments, population density, electric power infrastructure, crop lands, pasture lands, roads, railways, and navigable waterways. The original HFP 1993 dataset was published by Venter and colleagues [71], and these data have recently been updated and analyzed [38]. To compare with the LAP scores, we took the $1 \times 1$ km$^2$ site assessment of HFP at the exact position of the LAP view (only the $1 \times 1$ km$^2$-assessed degradation HFP score is compared with the landscape view LAP conservation index result). This, of course, gives a space-limited indication of the actual view since the extent of the view varies in every LAP site.

## 3. Results

A total of 50 sites were surveyed, although we originally envisioned to complete at least 80. The reason for this discrepancy relates to the difficulty of encountering actually suitable and representative viewpoints during the survey road trip. In fact, in such humid high-forest conditions, viewpoints are rather scarce; they are often advertised ("miradores") as the location of road stops and restaurants (but obviously, not always providing representative views). Each site was given a primary number code and named, i.e., a simple name inspired by the site's proximity to settlements or other features (Appendix A, Table A1). The route during the road trip and the selected sites are mapped (Figure 3), assessed based on the LAP conservation index and categorized into four landscape types (Figure 4).

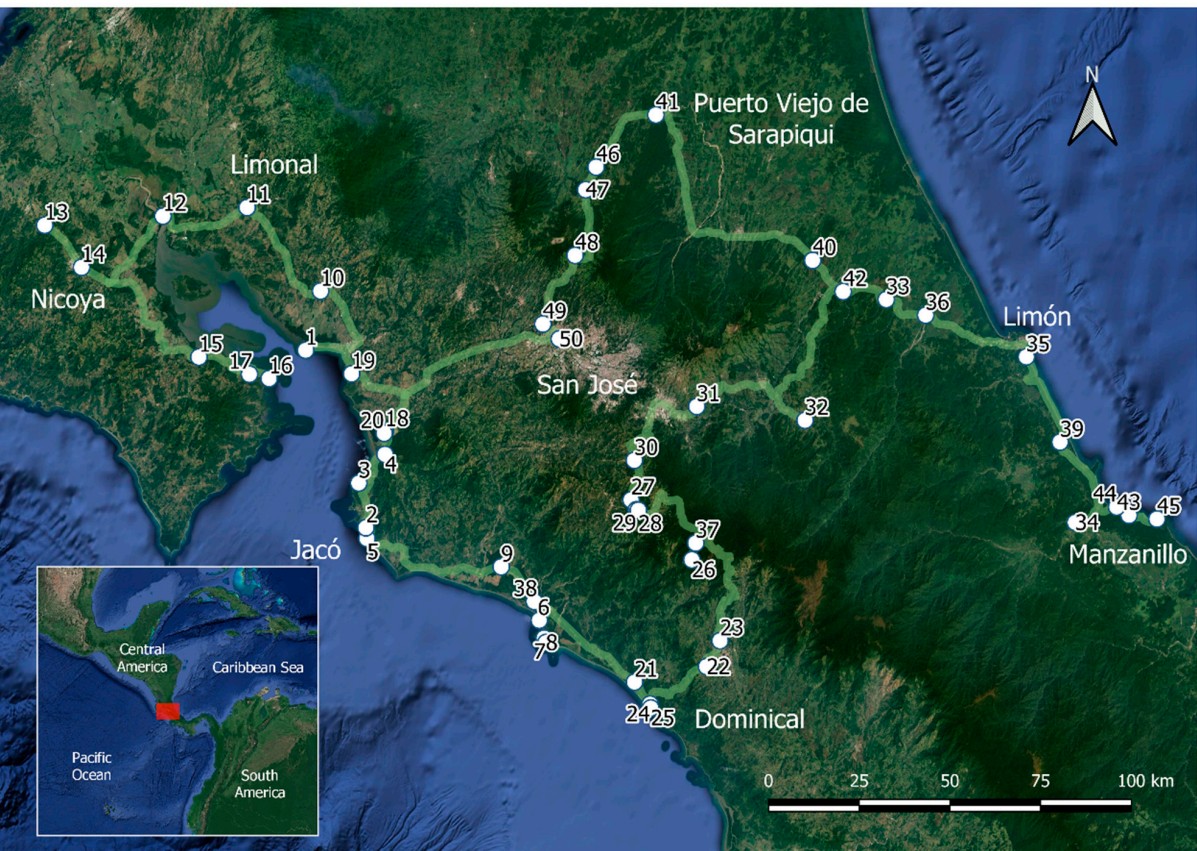

**Figure 3.** LAP sites indicated with white dots (number codes as in Appendix A, Tables A1 and A2). Route taken on the sampling survey expedition shown by light green line.

Due mainly to poor weather conditions, most assessments were completed at low elevations; less than 10% of the sites were above an elevation of 500 m (Figure 5). However, although it was not pre-planned, a large number of sites were assessed as being in moderate condition, with a wide spread across both degraded (poor–bad) and favorable conditions (good–excellent).

A general typology of the assessed landscape views was defined after the survey was completed. Based on the dominating geographical and characteristic landscape features at the assessed viewpoints, four generic landscape types were defined for descriptive comparisons and data presentation:

- Coastal (C): immediate contact with and dominance by the coastline;
- Mountain (M): high-relief landscapes, dominated by montane conditions;
- Flatland (F): low-relief landscapes, rolling hills, plains or plateaus;
- Urban/peri-urban (U): dominated by buildings in the immediate vicinity, within or next to settlements.

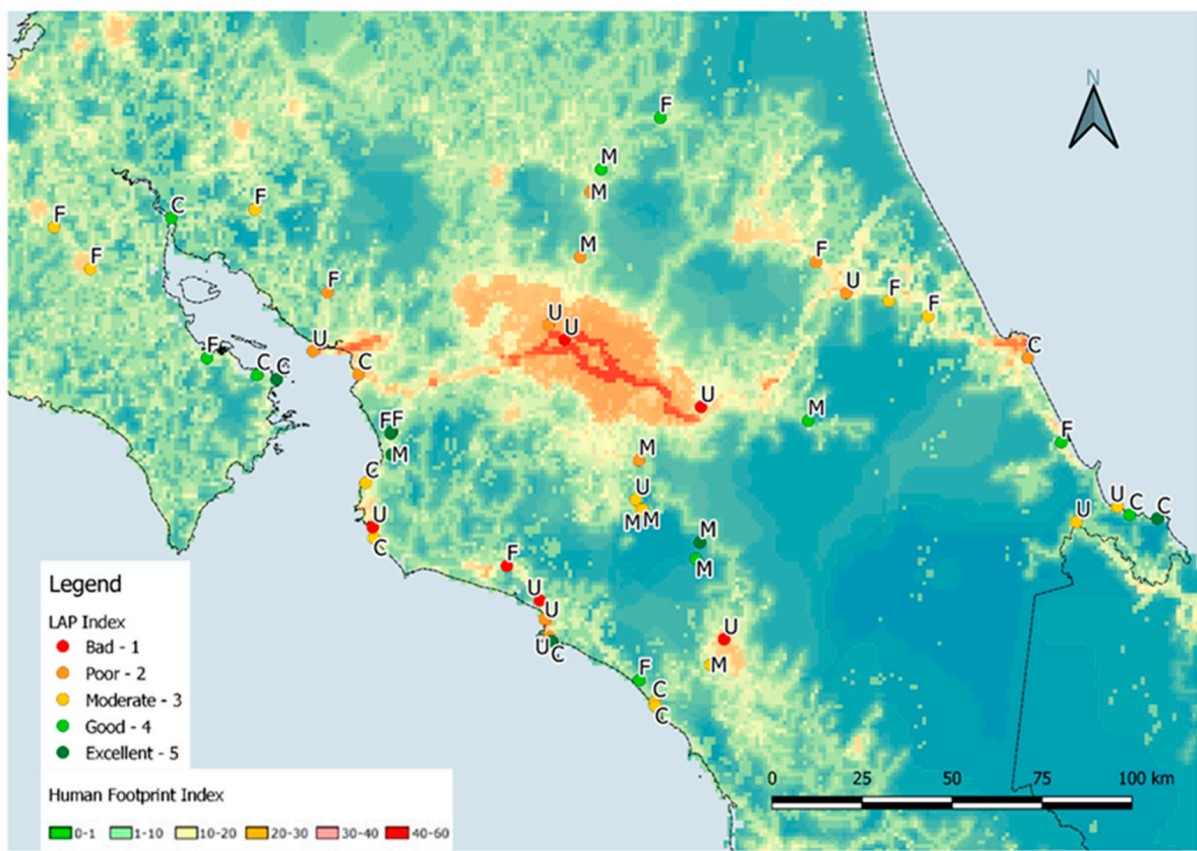

**Figure 4.** All surveyed sites overlain on the Human Footprint degradation assessment (warmer colors show anthropogenically degraded areas). Each point shows the LAP assessment index in five classes and the general landscape classification in four types (C = coastal, M = mountain, F = flatland, U = urban/peri-urban). The scale and map area in both maps is identical for comparative purposes.

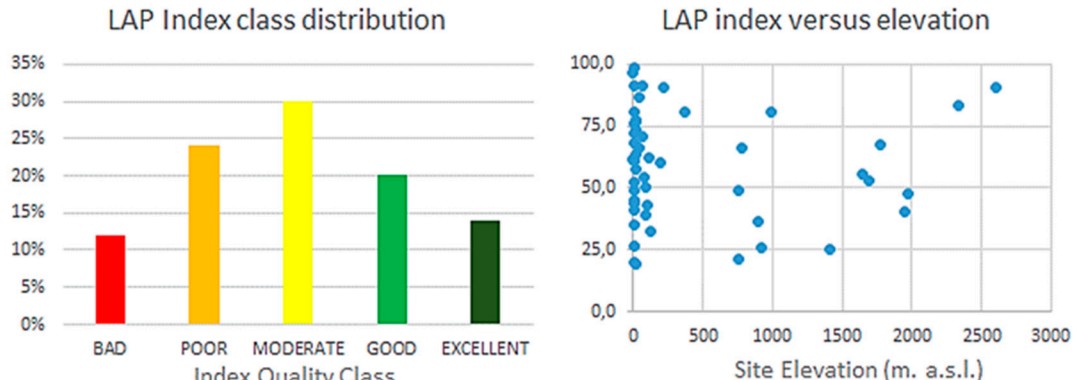

**Figure 5.** General results and distributions of the LAP survey. **Left**: Summary of final LAP index class assessments. **Right**: LAP index scores and elevation; no statistical significance existed for the relation of LAP (*Y*-axis) versus elevation (*X*-axis).

The distribution of scores per type are presented in Table 1; the urban and peri-urban landscape types differ markedly from all others. Of the 15 metrics, a handful were often not scored by the assessors (Table 2). The least scored metrics were: Hydrological Alteration (often hard to see water courses in the densely wooded landscape); Livestock Grazing (difficult to assess relative strength of negative impacts); Shorelines/Riparian areas (often hard to see water courses in the densely wooded landscape); and Agriculture (often missing from semi-natural and wilderness areas). It goes without saying that in some cases,

metrics may not be scored simply due to a lack of confidence in the particular situation (see discussion).

**Table 1.** Descriptive statistics for the raw LAP scores per type.

| Landscape Type | N | Mean | Std. Deviation | Std. Error | 95% Confidence Interval for Mean | | Minimum | Maximum |
|---|---|---|---|---|---|---|---|---|
| | | | | | Lower Boundary | Upper Boundary | | |
| Coastal | 12 | 70.223633 | 20.0008645 | 5.7737523 | 57.515690 | 82.931576 | 35.0000 | 98.3333 |
| Flatland | 14 | 61.358500 | 20.3314264 | 5.4338023 | 49.619484 | 73.097516 | 19.2308 | 91.3333 |
| Mountain | 11 | 68.125200 | 18.3823473 | 5.5424862 | 55.775771 | 80.474629 | 40.0000 | 90.7692 |
| Urban/Peri-Urban | 13 | 38.650700 | 14.5941715 | 4.0476949 | 29.831530 | 47.469870 | 20.0000 | 66.0000 |
| Total | 50 | 59.070778 | 21.9450071 | 3.1034927 | 52.834076 | 65.307480 | 19.2308 | 98.3333 |

**Table 2.** Metric mean score ± standard error (number of cases scored in parentheses) per each metric. The shaded have highest (lighter shade) and lowest (darker shade) values. Note that in some landscape views, the assessors chose not to score certain metrics, either because they were not perceivable or the assessor did not have enough evidence to score effectively; this is given in the last column.

| LAP Metric | Landscape Type | | | | Metrics Scored |
|---|---|---|---|---|---|
| | Coastal | Mountain | Flatland | Urban/Peri-Urban | |
| Land Use Pattern | 7.17 ± 0.716 (12) | 6.64 ± 0.62 (11) | 7.071 ± 0.45 (14) | 3.77 ± 0.57 (13) | 50 |
| Vegetation | 6.92 ± 0.723 (12) | 6.27 ± 0.76 (11) | 5.714 ± 0.56 (14) | 3.77 ± 0.52 (13) | 50 |
| Flora | 6.5 ± 0.821 (12) | 4.91 ± 0.79 (11) | 4.929 ± 0.6 (14) | 3.08 ± 0.58 (13) | 50 |
| Road Network | 6.75 ± 0.676 (12) | 6.18 ± 0.6 (11) | 6.308 ± 0.62 (13) | 3.31 ± 0.54 (13) | 49 |
| Modern Antropogenic Interference | 6.58 ± 0.633 (12) | 6.73 ± 0.56 (11) | 6.357 ± 0.58 (14) | 3.46 ± 0.43 (13) | 50 |
| Pollution, Garbage and Debris | 8 ± 0.59 (12) | 8.8 ± 0.51 (10) | 7.75 ± 0.73 (12) | 6.33 ± 0.61 (12) | 46 |
| Agriculture | 7.33 ± 1.229 (6) | 7.11 ± 0.59 (9) | 5.083 ± 0.75 (12) | 3.83 ± 1.35 (6) | 33 |
| Livestock Grazing | 6 ± (1) | 5.88 ± 0.77 (8) | 5.5 ± 0.76 (8) | 5.5 ± 2.5 (2) | 19 |
| Hydorological Alternation | 6.6 ± 1.077 (5) | 8.25 ± 1.11 (4) | 9 ± 0.32 (5) | 6 ± (1) | 15 |
| Shorelines and/or Riparian Conditions | 6.45 ± 0.755 (11) | 6.75 ± 1.6 (4) | 5.8 ± 1.36 (5) | 4 ± 0.32 (5) | 25 |
| Soundscape Quality | 6.08 ± 0.892 (12) | 7.1 ± 0.74 (10) | 5.615 ± 0.67 (13) | 2.46 ± 0.42 (13) | 48 |
| Landscape Attractiveness | 8.5 ± 0.5 (12) | 8.36 ± 0.75 (11) | 6.714 ± 0.73 (14) | 5.77 ± 0.74 (13) | 50 |
| Smellscape Pleasentness | 7.33 ± 1.067 (9) | 9 ± 0.6 (8) | 6.571 ± 1.51 (7) | 2.67 ± 0.44 (9) | 33 |
| Wildlife and Wildlife Habitat | 8.44 ± 0.852 (9) | 6.57 ± 1.51 (7) | 6.667 ± 1.67 (6) | 3.5 ± 0.67 (10) | 32 |
| Buildings | 6.33 ± 0.62 (12) | 6.18 ± 0.55 (11) | 6.25 ± 0.81 (12) | 3.67 ± 0.38 (12) | 47 |

As is evident in Table 3, and according to an ANOVA test of the means, the urban/peri-urban type of landscape received statistically significant, lower LAP scores than the rest of the types. Based on Levene's test, the standard deviation of the types was not statistically different, and thus the assumption about the difference of the types is valid (Table 3, Figure 6).

**Table 3.** ANOVA test of means with relation to landscape types.

| | Sum of Squares | df | Mean Square | F | Sig. |
|---|---|---|---|---|---|
| Between Groups | 7888.448 | 3 | 2629.483 | 7.700 | 0.000 |
| Within Groups | 15,709.135 | 46 | 341.503 | | |
| Total | 23,597.583 | 49 | | | |

| Test of Homogeneity of Variances | | | | |
|---|---|---|---|---|
| LAP | | | | |
| Levene Statistic | df1 | df2 | Sig. | |
| 0.444 | 3 | 46 | 0.723 | |

**Figure 6.** Mean LAP index scores per landscape type category.

From LAP's 15 metrics, Livestock Grazing and Hydrological Alternation are not correlated to the other metrics, with the exception of Land Use Pattern (See Appendix A, Table A3). Those two metrics were also the least-completed metrics through the 50 surveys (Table 2).

Principal Component Analysis (PCA) was used in order to identify the underlying relations of the metrics, e.g., typological, and to further assess which metrics were dominating the variance of the cumulative LAP score, either in a negative or positive manner. PCA was carried out using standardized metric values of LAP. Standardization was carried out by replacing the missing values with 5.5 and then setting the value 5.5 as 0, and setting the value 10 as 1 on the one end and the value 1 as −1 on the other. The horizontal axis of the PCA accounted for 64.6% of the total variation, whereas the second axis added a mere 6.9%. According to the arrangement of the 50 samples, the horizontal axis was evidently discriminating amongst the samples according to their LAP classification (Figure 7) and not according to the landscape type category; this provides evidence for validating the use of the LAP in very different landscape types. Furthermore, the first axis was most related to the values of Wildlife and Wildlife Habitat, Flora, Soundscape Quality, and, to a lesser extent, Vegetation and Smellscape Pleasantness (Table 4). The metrics of Wildlife and Wildlife Habitat, Flora, and Vegetation presented a high degree of concordance. The PCA shows that it is a multiparameter application, as a few metrics do not dominate.

One of the most challenging aspects of landscape view assessments is validating the index, i.e., finding proof from background conditions or other metrics that the index is objectively providing an accurate and consistent assessment. As previously mentioned, this could not be part of this rapidly executed survey; however, insights towards validation were explored by comparing the Human Footprint index (HFP) to LAP. HFP provides a value for the degree of degradation per square kilometer patch. We compare the HFP at the viewpoint position of the LAP. A simple correlation shows that a significant relationship was found; correlation coefficients whose magnitude are greater the 0.7 indicate variables which can be considered moderately to well correlated (Figure 8).

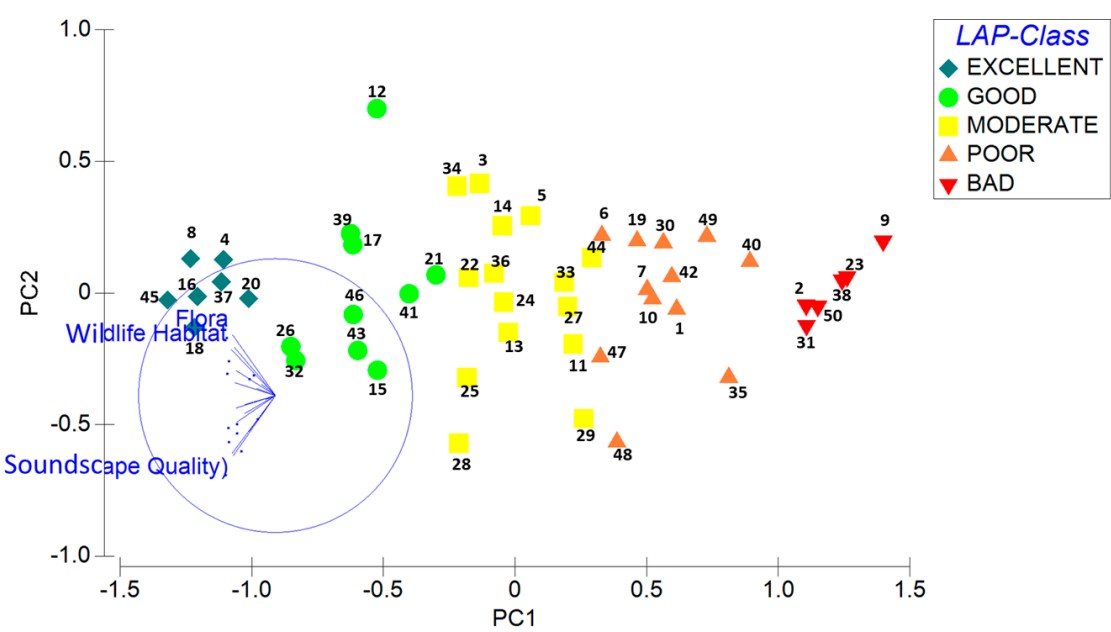

**Figure 7.** PCA plot for standardized metric values of LAP (numbers refer to each site).

**Table 4.** PCA eigenvalues and eigenvectors.

| Eigenvalues | | | |
| --- | --- | --- | --- |
| **PC** | **Eigenvalues** | **%Variation** | **Cum.%Variation** |
| 1 | 0.549 | 64.6 | 64.6 |
| 2 | 0.058 | 6.9 | 71.5 |
| 3 | 0.046 | 5.5 | 77.0 |
| 4 | 0.033 | 3.9 | 80.9 |
| 5 | 0.032 | 3.7 | 84.7 |

| Eigenvectors | | | |
| --- | --- | --- | --- |
| **(Coefficients in the Linear Combinations of Variables Making Up PCs)** | | | |
| **Variable** | **PC1** | **PC2** | **PC3** |
| Land Use Pattern | −0.286 | 0.182 | −0.128 |
| Vegetation | −0.301 | 0.354 | 0.054 |
| Flora | −0.314 | 0.444 | −0.017 |
| Road Network | −0.296 | 0.093 | −0.167 |
| Modern Antropogenic Interference | −0.286 | −0.197 | −0.074 |
| Pollution, Garbage & Debris | −0.195 | −0.265 | 0.474 |
| Agriculture | −0.226 | −0.066 | 0.469 |
| Livestock Grazing | −0.074 | −0.030 | 0.225 |
| Hydorological Alternation | −0.102 | 0.081 | 0.001 |
| Shorelines and/or Riparian Cond. | −0.136 | 0.052 | −0.226 |
| Soundscape Quality | −0.318 | −0.423 | −0.310 |
| Landscape Attractiveness | −0.289 | −0.092 | 0.473 |
| Smellscape Pleasentness | −0.308 | −0.439 | −0.255 |
| Wildlife and Wildlife Habitat | −0.325 | 0.339 | −0.075 |
| Buildings | −0.225 | −0.131 | −0.103 |

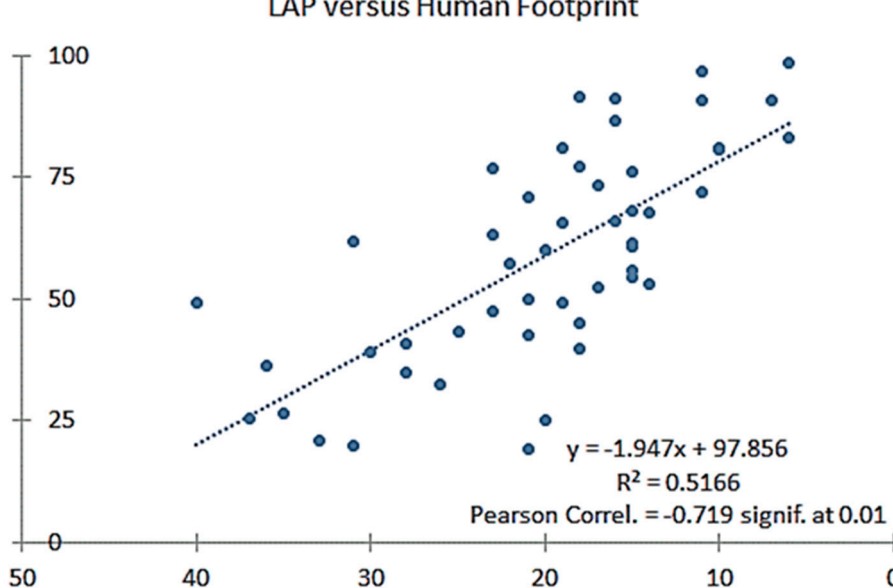

**Figure 8.** Correlation between the two indices, LAP score on Y-Axis and Human Footprint (HFP) on X-Axis. Note that the score gradient is reversed in the two indices: LAP scores of "0" mean degraded landscapes and "100" mean excellent ones; in the HFP, "0" stands for excellent (no impact) and "50" for the most degraded. Pearson correlation calculates the effect of change in one variable when the other variable changes, from moderately to well correlated in this case.

## 4. Discussion

### 4.1. Interpreting Anthropogenic Pressures on Landscapes

Our landscape assessments using the LAP correlate with the Human Footprint (HFP), as would be expected. Based on these results, it is our opinion that the multi-faceted landscape assessment provided by the LAP adequately addressed a variety of landscape types (Figure 9). This is an important result since the original LAP was originally developed in more northern-temperate and Mediterranean-climate regions. As expressed by Rapport [62], in terms of the patterns of anthropogenic disturbance, "natural systems, despite their diversity, respond to stress in similar ways". For example, it is usually obvious to the assessor that monocultures and urban sprawl affect biodiversity and the overall aesthetic landscape quality. Some of LAP's metrics focus on more cryptic details, including livestock grazing, flora impoverishment and wildlife habitat degradation, which are more difficult to consistently judge with precision solely from a specific landscape view. Utilizing several metrics may sometimes have an additive effect that contributes to a holistic assessment. In terms of wildlife richness, the general notion stands in tropical Mesoamerica: the more complex the natural vegetation and the larger the natural patches, the more species and human-intolerant "specialists" can thrive [19,64,72]. In general, a more natural landscape condition is also usually more aesthetically attractive [2]. The natural or semi-natural landscapes simply produced high scores on all of LAP's metrics in our study.

Despite Costa Rica's extensive protected areas, many landscapes outside of them suffer from land-use changes and degradation [22,67,73]. Our survey of 50 landscapes confirmed this, especially since many assessed sites were often near major roadways or near urban and touristic areas. There is also a degradation of cultural ecosystem services in the expanding number of "new" intensely managed landscapes; these may affect the interests of economically valuable wildlife tourism [74–76]. Expanding "nontraditional agricultural export crops" are especially notorious for serious degradation at the landscape level, along with negative social impacts as well [77,78]. If more intensely-modified landscapes regarding both monocultures and building sprawl are allowed to spread unchecked, we should expect widespread degradation that will affect both biodiversity and local society.

Landscape degradation trends such as these are common in the Latin American tropics (e.g., [9,38]).

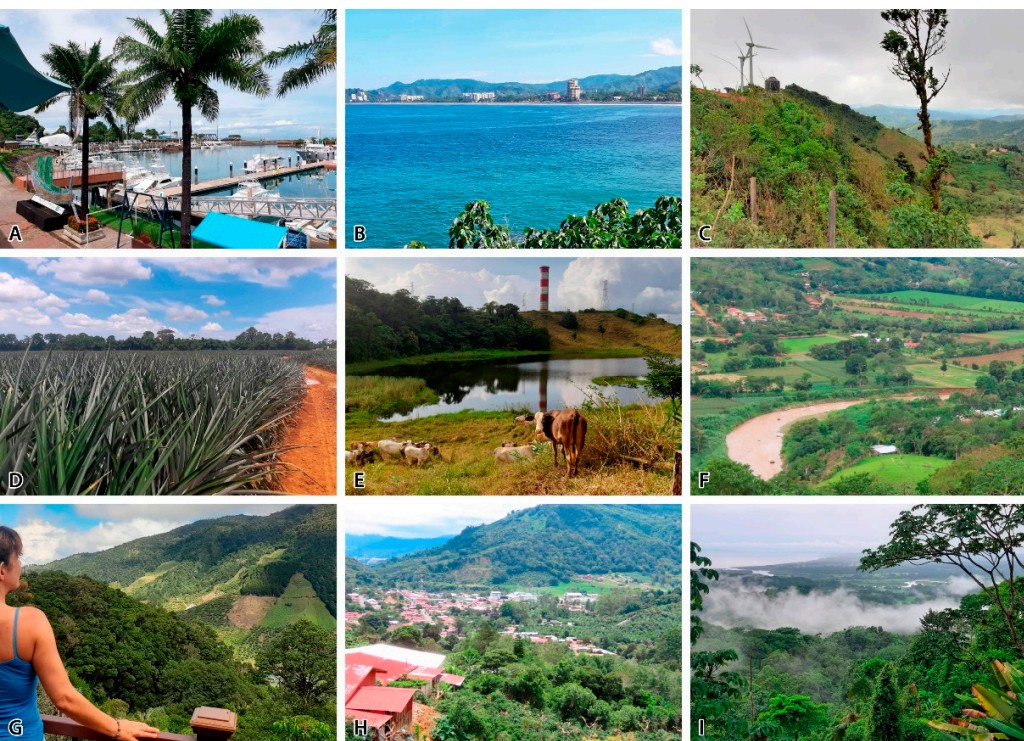

**Figure 9.** Sites with typical landscape-scale pressures and their five-class assessments applying LAP: (**A**) new tourist development (Marina Quepos, site 6: Poor); (**B**) coastline tourism (Jacó, Site 5; Moderate); (**C**) recent agro-pastoral expansion, new roads, and wind farm (Cristo Rey Desamparados, near Empalme, Site 30: Poor); (**D**) pineapple monocultures with remnant riparian forests (near Siquirres, Site 40: Poor); (**E**) overgrazing and artificial structures marring the horizon (Laguna Maria Aguilar near Pao Vulcano, site 47: Poor); (**F**) agricultural intensification and sprawl with riparian zone degradation (near San Isidro de General, site 40: Poor); (**G**) upland small-scale agriculture and regenerating rangelands (Savegre headwaters, San Gerardo de Dota, site 26: Good); (**H**) traditional multi-use coffee-growing landscape (near San Rafael de Dota, site 29; Moderate); (**I**) regenerating heterogeneous semi-natural landscape (Tarcoles delta from Carara National Park, Site 4: Excellent). All photos by V. Vlami and S. Zogaris during the survey.

LAP's 15 metrics were effective in providing an index for landscape quality showing a gradient of degradation in response to modern anthropogenic pressures. Despite these initial positive survey results, the issue of developing the best metrics for a robust index still requires thorough validation and more testing. This experience with the LAP in Costa Rica gives breath to a discussion on various aspects of the method used, including the onsite scoring of the index and the usefulness of onsite assessments under humid tropical and densely forested conditions.

### 4.2. Landscape Metrics Rating Insights and Challenges

In an effort towards explaining the results and gaining insights from the protocol's application, the rationale for using the LAP is discussed here. The LAP evaluates (i.e. rates) the status of the landscape's integrity, which should incorporate various measures of landscape-system conditions in order to document and quantify losses in value due to negative anthropogenic impacts [61]. The concept of rating in the LAP is based on reference conditions; these are baseline conditions described in the scoring guidance narrative at a rating scale of "10", i.e., the state of an "excellent" condition for each metric (see Appendix A, Figure A1). Identifying reference conditions is therefore a key to measuring

landscape quality. References and baseline classification boundaries should be informed by knowledge of natural history and landscape history, otherwise one may fall victim to the shifting baseline syndrome, i.e., when human perceptions of changed conditions may misguide baselines of the natural or optimal ecological state [79]. Rating landscape conditions should also assist in both developing aptitude and designing further inquiries towards landscape quality assessment (i.e., developing landscape literacy and diagnosis).

Reliably rating the visual or perceptual qualities of landscapes has a rather recent history of method standardization [35,80]. The numerical rating scale that the LAP uses is similar to the Stream Visual Assessment Protocol (SVAP) format [81], continuing the practice of rapid bioassessment protocols [51,55]. It provides the 10-to-0 rating scale (an 11-point rating system) currently widely used in healthcare and biotic assessment [82]. Rating should help the assessor distinguish signals that reveal relevant content from the "noise" in a system. Yet, rating landscape metrics is an exercise in quantifying the assessor's perceptions, and is prone to subjectivity. It has been said that reference baselines are "social constructs" and the baselines could arbitrarily vary [1]. Controlling for such subjectivity is based on metric guidance structure. The LAP promotes and permits leaving some metrics "unscored" and this may help to point out which of these evaluative criteria, which metrics, could not be judged accurately, could not be perceived consistently in the views, or may produce evaluation difficulties (see Section 3, Table 2).

The LAP favors the 10-to-0 scale but also utilizes a simplified 5-class-scale summary of conditions to display the summary quality condition (the index result from excellent to bad). The 5-class scale is widely used in policy-relevant applications such as Europe's Water Framework Directive assessment and monitoring applications. However, a longer scale (i.e., *contra* to 1–5) shows more variety of differentiation, and in this way the 11-point scale should increase variability and also precision. With 11 rating options, the 10-to-0 scale also gives a true average rating (i.e., the number 5), indicating when a metric condition was neither favorable nor unfavorable. So, we believe it is correct to provide a longer scale for the initial onsite assessment and summarize the overall results using a simpler five-class scale. For this application in Costa Rica, we think the spread of scores for the five-class scale provided satisfactory communication of the assessed landscape status, as well as the cartographic visualization.

We must reiterate here that effective scoring techniques require practical experience and training [58]. In scoring a 10-to-0 scale, one must always weigh items carefully, and this should require lots of practical experience since sloppiness may easily and haphazardly "creep into" the assessment process. Each participant may also have different "personal" benchmarks for scoring, and this requires careful streamlining through training or using the consensus method, as we did in this application (i.e., scoring based on what two people think instead of one). Scoring is also prone to technique rituals (i.e., rules of thumb). The following remark attributed to the athlete Kyle Maynard refers to ranking personal physical performance on a 10-to-0 scale: "Removing 7 gives you a far better signal on everything. 7 is the most common default and tells you little; conversely, 6 is barely passing and 8 is a strong endorsement" (Ferriss, T. pers. communication, 2021). Rules of thumb may assist in training schemes and may help in streamlining the assessor's rating consistency.

To explore potential unmet needs or any shortcomings in the LAP rating mechanism one should include an exploration of the metrics individually (and then as groups of measures that are related); for example, sets such as diversity and integrity gradients, gradients of human use such as urbanization, etc. There is a need to better understand which of the metrics of the LAP are useful in determining what the landscape condition is and what is causing that condition (J.R. Karr, personal communication). The final LAP index obviously attempts a generalization: "competing" metrics may eclipse others [83], i.e., the sum of metrics may not have the desired positive additive effect. Future research may need to be conducted for specifying certain metrics or re-calibrating class boundaries. Some metrics may have to be re-labeled as optional (e.g., the Smellscape Pleasantness metric was within the five less-scored metrics in this application in Costa Rica; should

it be downgraded to an optional metric?). The development of reliable metrics for such "ecological" assessments is an ongoing process [84], and building robust environmental indicators for ecosystem and landscape quality is still an area of active study [85]. Finally, some rapid assessment methods have fared well without any adaptations to "better" conform to local bioregional realms (i.e., they transfer across different continents well), and their standard use in many different jurisdictions has been useful [55,86].

*4.3. Identifying "Traditional Cultural Landscapes"*

In Latin America, the idea of cultural landscapes is often expressed differently from that of the North American and European traditions [40,87]. To some degree, nearly all landscapes are cultural, but there are important distinctions. As coined in 1925 by the American geographer Carl Sauer, "a cultural landscape is transformed from a natural landscape by a local group. Culture is the agent, and the natural area is the medium. The cultural landscape is the result of that transformation" [3]. It is now widely understood that cultural landscapes are places where human action is displayed through the historical transformation of nature, but it takes time for cultural landscape patterns to develop and be sustained in an identifiable cultural landscape state. Schmitz, García, and Herrero-Jáuregui [88] summarized this well in 2021: "cultural landscapes are the result of social–ecological processes that have co-evolved throughout history, shaping high-value sustainable systems". The construction of unique landscapes full of history and cultural content, including intangible cultural values and distinct biocultural features, should be sought for and rated positively in visual landscape assessment. In Latin America, some of these landscapes are often called "traditional landscapes" [13]. People intuitively are attracted to the traditional cultural landscape image: the nostalgic notion of "the Costa Rica of yesterday" is fleetingly described sometimes in the tourism literature, for example. However, how can we consistently distinguish between a traditional cultural landscape and a recently degraded or recently regenerating human-modified landscape?

The basic premise here is that traditional cultural landscapes should be differentiated from modern, recently disturbed, human-modified landscapes. Landscape change is space-specific, and generalizations are difficult with the remarkable heterogeneity of landscape forms, especially in the humid tropics, where vegetation regeneration is very rapid [89,90]. Most discourse is rather arbitrary, and little work has been completed for providing a typology for cultural landscapes in the American tropics. In the conservation literature, an often and broadly used term is "working landscape", which obviously does not characterize them as cultural or traditional. Since a lot of landscape change has taken place relatively recently in much of tropical Latin America, it is sometimes difficult to distinguish between long-term subtle changes and rather recently "opened" and quickly regenerating landscapes. After political changes, extensive areas have seen forest regrowth throughout Costa Rica [65]. Much of the vegetation has regenerated back after the abandonment of small-scale agriculture, making visual onsite assessments difficult for the untrained eye (i.e., in the Nicoya peninsula; e.g., see comments in D.R Wallace's account [91]). This issue was plainly evident in our field experience with the LAP in Costa Rica, and further inquiry into interpreting the attributes of cultural landscapes should be investigated. In our opinion, the importance of cultural landscapes as designated conservation areas [29] and areas of cultural or scenic value also has potential the Latin American tropics. LAP contains metrics that are potentially useful for such a diagnosis at the screening level, i.e., the first tier of landscape conservation surveying [57]. It incorporates a rapid biodiversity assessment platform that provides important interpretation approaches, such as attention to small landscape elements [92], various aspects of human history, and cultural features, and is not dominated solely by evaluating scenic attractiveness in a landscape view.

*4.4. Recommendations*

Insights that may merit further inquiry emerge from our application. Landscape studies are critically important for conservation, especially in semi-natural and disturbed

areas and particularly outside of protected areas [3,10]. Efforts to assist landscapes and communities in peri-urban and "commodity production" landscapes should be actively investigated [93]. Our approach promotes an effort for "hands-on" participation within the framework of the wider promotion of landscape literacy. The LAP could be useful for landscape conservation because it can help build a wider sensitivity to the landscape scale. This kind of assessment is important for interdisciplinary approaches, including inventory and monitoring by various stakeholders. In recent years, many interdisciplinary and transdisciplinary initiatives have strived to employ landscapes in bold and ambitious ways within conservation-relevant research. There are important efforts to promote multi-functional agriculture that maintains agricultural productivity while simultaneously conserving biodiversity [21,72,93]. Several long-term studies in Costa Rica have shown that even minor improvements in agricultural practices can increase biodiversity and its benefits to local communities [19,94–97]. Utilizing landscape-scale approaches will continue to expand and widen conservation and management horizons.

One of the key challenges in addressing landscape change is acquiring an objective and shared understanding of landscapes within and among disciplines [56,98]. Landscape understanding is important for the multi-faceted interdisciplinary practice of conservation [10]. This ranges from urban planning issues to threats to species' survival over large areas [99]. It is our opinion that inventory and assessment initiatives must be the basis of a hierarchical development towards an integrated nature-culture heritage conservation that should culminate in effectively conserving landscapes (Figure 10). The LAP and other on-site assessment and participatory applications could help spread a common understanding of landscapes and landscape-scale problems.

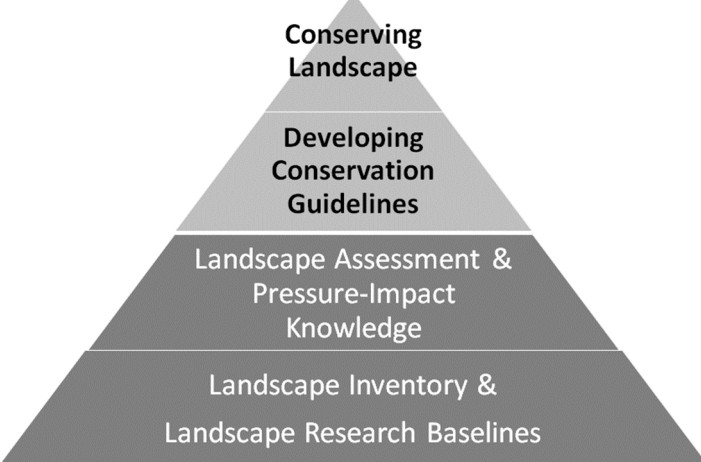

**Figure 10.** Why landscape assessment should be the basis for integrated nature-culture conservation. Assessment methods such as LAP act in the bottom two tiers of this symbolic hierarchical pyramid (inspired by Feinsinger 2001 [10]).

The insights gained from this application in Costa Rica and a review of the issue for the Latin American tropics lend support to the following recommendations:

- The use of the LAP as an onsite protocol for landscape view assessments has positive prospects in the tropics of Latin America and could be widely applied as a first-tier screening survey method. The original LAP scoring system provides a standard method and could be used widely without any changes or adaptations. Efforts to better adapt the LAP to regional conditions in the tropics may be investigated after in-depth inquiry. Tweaking the metrics, the numerical class-boundaries, and other aspects of the protocol is a natural evolution of a useful assessment protocol. However, this endeavor should only be started after much evidence is gathered and within a regional intercalibration or standardization process.

- LAP may combine well with other landscape assessment endeavors (e.g., HFP data); it can be used in "ground truthing" comparisons and in parallel with other approaches that assist in inventory and scoping evaluations of landscapes.
- The inquiry into landscape view assessments may help promote policy recommendations for the preservation and restoration of landscape quality, including a plea for the active promotion of landscape protection measures and restorations outside of protected areas. Economic incentives for landscape restoration may be further incorporated [100].
- Costa Rica and other Latin American countries should seek to pioneer actions to define protected area categories that include the landscapes outside of traditional park designations. A focus on traditional cultural landscapes should be promoted. LAP may help in inventory, assessment, and cartography, as well as in relevant public participation studies.
- Community and citizen participation is important; a key effort must be made to promote landscape literacy and public, student/youth, and resident minority participation. Landscape literacy develops from experience and the LAP provides an onsite tool. A participatory platform such as the LAP could be instrumental for engaging youth, students, locals, and visitors.

An important aspect of this sort of landscape-assessment inquiry is its potential for promoting field-based methods of education and public awareness. Field-based approaches to education have been widely expounded since the excellent manifesto of Lonergan and Andresen [101]; however, the development of "natural history" skills in this context is more challenging [10]. Natural history knowledge requires field work to build perceptions and aptitude. There is a conceptual common ground for visual and ecological landscape indicators [2], and students may be assisted by broadening multiple intelligences that are poorly developed in classroom education. In our experience, the LAP is very capable in providing a focused perspective and methodology to engage students, just as with other onsite rapid-bioassessment-style protocols, including SVAP [81] and QBR [86]. The LAP is very conducive to training and uptake by students in outdoor education and field course conditions [58]. Furthermore, the LAP may also be a tool to introduce aspects of indigenous knowledge within the landscape view assessments, since it could be used as an adaptive method in striving for a more inclusive and holistic diagnosis of landscapes [102]. The LAP could also assist in other interdisciplinary approaches, including methods of interviewing residents, to explore the landscape perceptions of local communities [103]. We feel that support for landscape literacy is one of the most important potential contributions of the LAP; this simple, holistic survey method may be able to promote landscape citizen science [104], providing various benefits towards effective landscape conservation advocacy.

## 5. Conclusions

To the best of our knowledge, our study is the first to explicitly examine the use of the original LAP method in a country-wide transect survey of the Latin American tropics. The LAP was effective in providing an index for landscape quality showing a gradient of responses to various forms of modern anthropogenic impact. The index results seemed to respond and document anthropogenic degradation fairly well, as the comparison to a remotely sensed index for land degradation showed. The LAP index consistently detected landscape-scale degradation near urban and peri-urban areas in particular. We recommend that the onsite method should be promoted and further applied. Besides the need for further validation research, the LAP also seems to be a good educational tool for landscape literacy and conservation promotion, and could also support and help supplement other landscape-scale assessment methods.

**Author Contributions:** Conceptualization, V.V. and S.Z.; methodology, S.Z., V.V. and C.M.B.; formal analysis S.Z. and V.V.; investigation, S.Z. and V.V.; resources, S.Z., C.M.B. and V.V.; data curation, V.V. and S.Z.; writing—original draft preparation, S.Z., V.V. and C.M.B.; writing—review and editing, S.Z., V.V. and C.M.B.; visualization, V.V. and S.Z.; project administration, V.V. and S.Z. All authors have read and agreed to the published version of the manuscript.

**Funding:** This research received no external funding.

**Institutional Review Board Statement:** Not applicable.

**Informed Consent Statement:** Not applicable.

**Data Availability Statement:** Data is available from the corresponding author upon request.

**Acknowledgments:** The authors wish to thank James R. Karr for sharing ideas for the LAP development, his discussions have been extremely helpful. We thank Yorgos Chatzinikolaou, who provided substantial assistance and advice for the statistical analysis; Christos Kontopoulos and Aris Vidalis for assistance with cartographical and graphic design support.

**Conflicts of Interest:** The authors declare no conflict of interest.

## Appendix A

| 1. Land Use Pattern | | | | | | | | | | |
|---|---|---|---|---|---|---|---|---|---|---|
| Original landforms, vegetation and cultural landscapes. Traditional elements and natural features intact (traditional architecture, etc). | | Original landforms dominate. Slight modern changes and breaks in patterns of traditional land uses. | | Moderately degraded. Some signs of changes in traditional land use and bio-physical patterns. | | Very few natural and traditional cultural patterns. Disorder and disharmony, notable degradation. Recent changes may be evident. | | No natural and traditional cultural features. Modern elements and features dominate. Multiple recent changes and disorder dominate. | | |
| 10 | 9 | 8 | 7 | 6 | 5 | 4 | 3 | 2 | 1 | 0 |
| **2. Vegetation** | | | | | | | | | | |
| Natural vegetation or centuries-old traditional culturally-modified vegetation. Reference conditions as defined by local natural history (i.e. not necessarily climax vegetation). | | >70% natural vegetation or traditional cultural landscape vegetation. Slight modification from reference conditions. | | < 50% natural vegetation or cultural landscape features. Moderately degraded. | | <30% vegetation or cultural landscape vegetation. Widespread degradation and/or modern pressures affect vegetation patterns. | | No natural vegetation; no centuries-old culturally-modified vegetation types. Totally degraded or artificial (i.e. recent plantations, etc.) | | |
| 10 | 9 | 8 | 7 | 6 | 5 | 4 | 3 | 2 | 1 | 0 |
| **3. Flora** | | | | | | | | | | |
| Native flora, natural or near-natural assemblages; or semi-natural centuries-old traditional landscape. Habitat types/plant communities related to the reference plant-association patterns. | | Slight degradation due to anthropogenic influence; otherwise natural floral assemblages cover >70%. Natural flora dominant. | | Moderately impacted. Human-tolerant species present; low species diversity due to disturbance (e.g. wild-fire). | | Totally degraded plant communities by human-induced changes. Non-native species may dominate. Very degraded by anthropogenic pressures, recent modern changes affecting flora. | | | | | | |
| 10 | 9 | 8 | 7 | 6 | 5 | 4 | 3 | 2 | 1 | 0 |
| **4. Road Network** | | | | | | | | | | |
| No modern paved roads; only traditional trails, small unpaved tracks. (In traditional high-integrity settled landscapes included small areas of paved roads, cobbled and other very old routes evident). | | Only small dirt roads present. Very low density road network. Roads narrow. > 70% of view has no roads present. (No urban, peri-urban densely roaded areas) | | Few roads; no large highways or many paved roads but paved road network covers several parts of landscape. (Urban, peri-urban areas). | | Road network nearly dominant. Evidence of bad engineering practices. Severe habitat fragmentation evident. (Sprawling urban, peri-urban areas) | | Road network dominant in landscape. Or, even when few roads present, signs of widespread road degradation (visible erosion, lands-slides, fragmentation, etc). | | |
| 10 | 9 | 8 | 7 | 6 | 5 | 4 | 3 | 2 | 1 | 0 |
| **5. Modern Anthropogenic Interference** | | | | | | | | | | |
| All human-built structures are traditional; no urban, industrial or other sprawl. No structures or buildings breaking the horizon (no wind farms, electricity networks, telecommunications, etc). Rural or natural scene dominates. | | Slight influence of human-made structures (very few utility poles, isolated or single structures). No structures or buildings breaking the horizon (pylons, electric wires, wind farms etc). Rural or natural scene dominates. | | Modern anthropogenic structures immediately apparent. A few structures slightly breaking the horizon at least in one position on the horizon. Rural/ natural environment still dominates. (In urban areas this state is near-reference, i.e. much green-space, traditional architecture dominates). | | High anthropogenic structures evident in some areas (electric wires, tall structures). Modern buildings and high structures break the horizon at several (2-5) places on the horizon. Some structures far away, e.g. wind farms at a distance. (Urban or peri-urban environment show disorder and loss of integrity). | | Totally altered by modern anthropogenic structures and/or recent construction. Many structures such as new buildings and linear structures breaking the horizon at several places (5+) on the horizon. Multiple changes show dominant anthropogenic fragmentation. Wind farms dominate on nearby ridgelines. (Urban areas with no natural areas) | | |
| 10 | 9 | 8 | 7 | 6 | 5 | 4 | 3 | 2 | 1 | 0 |

**Figure A1.** *Cont.*

## 6. Pollution, Garbage & Debris

| No garbage and no heavy construction site debris or other anthropogenic debris in sight. | | Very small quantities of garbage (light pieces scattered). Slightly altered conditions (e.g. due to older dumping activity, now totally overgrown (very localized)). | | Noticeable scattered trash. and/or some scattered construction-site debris evident. Slightly altered conditions due to general disorder (localized dumping/in-filling etc.). | | Several areas of garbage dumped in sight and/or large quantities of debris. Extensive infilling may be apparent (in-filled wetlands, grasslands, vacant lots). Water pollution evident. | | Severe garbage and trash dumping in sight. Much of trash and debris dumped in large quantities (10+ truckloads). Also may include large mounds of debris or other forms of pollution. | | |
|:--:|:--:|:--:|:--:|:--:|:--:|:--:|:--:|:--:|:--:|:--:|
| 10 | 9 | 8 | 7 | 6 | 5 | 4 | 3 | 2 | 1 | 0 |

## 7. Agriculture

| No industrial agriculture/monocultures present. If agriculture present, only traditional forms exist in relatively small scale; rather small parcels. High nature value farming practices evident (natural breaks, hedgerows, much habitat for wildlife and insect life). | | Biodiversity rich agricultural lands with high nature value farming practices. Less than <30% of landscape under monocultures; varied forms of traditional agriculture and small-scale holdings dominate. | | Moderate agriculture impact. Monocultures present. Low nature value farming practices evident, but a good balance of varied agricultural practices with much "nature" present on farms. | | Degraded by agriculture. At least 50% of landscape in modern monoculture. Many associated modern infrastructure elements. Intensive farming (e.g. greenhouses) and industrial farms. | | Excessive intensive agriculture or poorly placed crops; dominant chemically-supported industrial agriculture. Monocultures dominate and farming structures (greenhouses etc.) common. Nearly no natural habitat present on farms. | | |
|:--:|:--:|:--:|:--:|:--:|:--:|:--:|:--:|:--:|:--:|:--:|
| 10 | 9 | 8 | 7 | 6 | 5 | 4 | 3 | 2 | 1 | 0 |

## 8. Livestock Grazing

| Livestock grazing conditions natural or sustaining traditional multifunctional landscapes (no recent abandonment evidenced). If no grazing apparent; wildlife grazing evident. | | Slight or some evidence of negative grazing impacts. Some overgrazing or abandonment may be evidenced. Otherwise impacts of grazing not severely detrimental to reference biodiversity patterns that would be present in optimal conditions of grazing or browsing pressures. | | Moderate evidence of negative grazing impacts that may show recent vegetation degeneration or degrading certain habitats (i.e. forest regeneration). Strong grazing impact; or conversely, recent "total abandonment" and homogenization. | | Overgrazed conditions. Noticeable vegetation degeneration process (severe changes in ecological succession pattern). Severe erosion from trampling. Stunted shrub and tree growth. Grass and herb scarcity. Livestock droppings and trails in abundance. Grazing after fire/logging and associated with vegetation clearing.  (This assessment of "overgrazed status" varies with respect to vegetation type and cultural traditional land-uses; must have good natural history knowledge to interpret the degree of overgrazing present). | | | | | |
|:--:|:--:|:--:|:--:|:--:|:--:|:--:|:--:|:--:|:--:|:--:|
| 10 | 9 | 8 | 7 | 6 | 5 | 4 | 3 | 2 | 1 | 0 |

## 9. Hydrologic Alteration

| All river and streams in apparently natural condition. No dams, no serious water withdrawals, no dikes or other structures affecting flow regime or limiting the stream access to the floodplain. Wetland conditions in natural or near-natural state. | | Slight changes to hydrology apparent. Withdrawals, although present, do not affect natural flow regime and/or available habitat for biota. Wetlands in good condition despite some alteration or human-induced changes. | | Development of landscape shows significant negative effects to flow regime exist. Moderate changes throughout river basin. Altered flows/water levels may be apparent. | | Degraded hydrology. Water withdrawals significantly affect flow regime and/or available habitat for biota. Artificial structures dominate near waterways and wetlands. | | Very degraded hydrology. Withdrawals, channelization or piping have caused complete alteration of flow regime; severely affecting aquatic habitats. Dams, embankments etc. severely degraded wetlands. | | |
|:--:|:--:|:--:|:--:|:--:|:--:|:--:|:--:|:--:|:--:|:--:|
| 10 | 9 | 8 | 7 | 6 | 5 | 4 | 3 | 2 | 1 | 0 |

## 10. Shorelines &/or Riparian conditions

| All shorelines and riparian zones natural. No roads, buildings, harbors, noticeable artificial structures. Coastal process and riparian areas formed and in natural processes. | | All shorelines in near-natural condition. Slight changes and localized degradation by a few roads, isolated buildings or other minor modern structures. Riparian areas mostly with natural vegetation. | | Moderate and noticeable change apparent (nearly 50% of shoreline/riparian zones altered). | | Degraded or altered. Most of the shorelines/and or riparian zones (>50%) altered or built-up by modern uses and infrastructures. Less than 30% natural vegetation present. | | Shorelines/and or riparian zones built up; altered by modern uses and infrastructures. No natural floodplains or riparian habitats, embankments may dominate. Only few tolerant and alien species and plantations on riparian zones. | | |
|:--:|:--:|:--:|:--:|:--:|:--:|:--:|:--:|:--:|:--:|:--:|
| 10 | 9 | 8 | 7 | 6 | 5 | 4 | 3 | 2 | 1 | 0 |

## 11. Soundscape Quality

| 100% natural and traditional sounds. dominate. No or very few artificial modern mechanical sounds. | | Nearly all near-natural and traditional sounds dominate. Slight mechanical sounds in distance (but, e.g. no frequent road noise). | | Small road and/or scattered agricultural noise break up natural or traditional cultural sounds. (Road noise in the distance). | | >70% modern anthropogenic sounds dominate. Heavy road noise. Other disturbing noises may come-and-go (e.g. overflying planes frequent). Some natural sounds apparent. | | 100% mechanical sounds dominate. Noises load and may be unpredictable. No or extremely few natural sounds apparent. | | |
|:--:|:--:|:--:|:--:|:--:|:--:|:--:|:--:|:--:|:--:|:--:|
| 10 | 9 | 8 | 7 | 6 | 5 | 4 | 3 | 2 | 1 | 0 |

## 12. Landscape Attractiveness

| Exceptionally attractive; richly varied; rare landscape. Exemplar natural or cultural features or elements. Outstanding scenic quality. | | High attractiveness; only slight conditions or elements impinging on natural/cultural aspects. Otherwise scenic and interesting. | | "Average attractiveness". Moderate natural elements, but some modern changes. Some degradation. | | "Unattractive". Degraded by human changes, not scenic in any way. | | "Ugly landscape". Drab and unattractive. Altered and degraded mostly by human interventions. | | |
|:--:|:--:|:--:|:--:|:--:|:--:|:--:|:--:|:--:|:--:|:--:|
| 10 | 9 | 8 | 7 | 6 | 5 | 4 | 3 | 2 | 1 | 0 |

## 13. Smellscape pleasantness (optional)

| No unpleasant smells; natural and culturally authentic smells dominate. | | | | Moderate artificial slightly unpleasant smell from human sources. | | Some unpleasant smells related to anthropogenic degradation. | | Very unpleasant smell from anthropogenic sources. | | |
|:--:|:--:|:--:|:--:|:--:|:--:|:--:|:--:|:--:|:--:|:--:|
| 10 | 9 | 8 | 7 | 6 | 5 | 4 | 3 | 2 | 1 | 0 |

**Figure A1.** *Cont.*

| 14. Wildlife and wildlife habitat | | | | | | | | | | |
|---|---|---|---|---|---|---|---|---|---|---|
| Wildlife hotspot and habitat-rich landscape. "Special habitats" present (e.g. wetlands or other scarce habitat types). Presence of species intolerant of urban or human-disturbed areas. High wildlife populations (specialist species of birds/insects) may be evident or presumed/or known to exist within landscape. | | Slightly disturbed but good conditions for wildlife present; intolerant and/or rare species known to be present. No or few domestic/feral/invasive species apparent. Some scarce "special habitats" and/or rather high habitat diversity (small wetlands, rare woods, grasslands etc) present. | | Moderate wildlife habitat values (far from what would be expected in natural conditions). No "special habitats" present. If "special habitats" present they are moderately degraded. (Urban conditions with rich wildlife habitats) | | Poor, altered wildlife habitat. No special conditions or refugia (no "special habitats" present). Very few wildlife species seen. Some wildlife may be present or their habitat potential present but an absence intolerant species. (But good urban wildlife areas ). | | Nearly no wildlife habitat present. No wildlife present (or only overflying and far from location of site assessment). Completely degraded, altered habitats and poor overall conditions for biodiversity. (Typical for many inner city urban areas or suburban areas with no wildlife-friendly parks or open spaces). | | |
| 10 | 9 | 8 | 7 | 6 | 5 | 4 | 3 | 2 | 1 | 0 |

| 15. Buildings | | | | | | | | | | |
|---|---|---|---|---|---|---|---|---|---|---|
| If outside defined settlement, modern buildings are only in defined legal area. If inside settlement, not illegal or unsightly modern buildings (i.e. in harmony, balance, order) and traditional authentic features very well preserved. High integrity, planning and order in urban and peri-urban environments. | | | | Moderate landscape degradation due to new buildings, but very little widespread sprawl effect (i.e. few scattered buildings). | | If outside defined settlement, scattered disorganized modern buildings and sprawl. Severe fragmentation and unsightly change. If inside settlements, illegal or unsightly elements dominate (i.e. disharmony, disorder, incompatible forms, lack of orderly building; no planning or restrictions enforced) | | | | |
| 10 | 9 | 8 | 7 | 6 | 5 | 4 | 3 | 2 | 1 | 0 |

**Figure A1.** The LAP scoring guidance sheet. Showing the guiding narrative and relevant score gradient (10 = reference conditions, i.e. "excellent" condition; 0 = poorest degraded quality, "bad" condition). This guidance sheet is used alongside the LAP field form's scoring card (Figure 1) to guide scoring of landscape views on site.

**Table A1.** Site dataset with location information (coordinates in WGS84—Decimal Degrees (DD).

| SITE # | Longitude (WGS84) | Latitude (WGS84) | Landscape Type | Elevation m.a.s.l. | Date | Site Name |
|---|---|---|---|---|---|---|
| 1 | −84,838632 | 9,973941 | Urban/Peri-Urban | 3 | 6.08.21 | Puntarenas |
| 2 | −84,630709 | 9,614198 | Urban/Peri-Urban | 11 | 8.08.21 | Vista Mar Jaco (near Hotel Del Mar) |
| 3 | −84,665684 | 9,705110 | Coastal | 1 | 8.08.21 | Playa Mantas |
| 4 | −84,605806 | 9,762929 | Mountain | 218 | 8.08.21 | TIKO Mirador Carara |
| 5 | −84,624773 | 9,592389 | Coastal | 14 | 9.08.21 | Mirador Jaco |
| 6 | −84,166989 | 9,426671 | Urban/Peri-Urban | 5 | 9.08.21 | Marina Quepos |
| 7 | −84,149293 | 9,390509 | Urban/Peri-Urban | 10 | 9.08.21 | Manual Antonio Tourist Beach |
| 8 | −84,143438 | 9,381608 | Coastal | 1 | 9.08.21 | Manual Antonio NP Beach |
| 9 | −84,279056 | 9,535017 | Flatland | 21 | 9.08.21 | Rio Palo Seco |
| 10 | −84,820662 | 10,094085 | Flatland | 96 | 10.08.21 | Rancho Grande Gasolinero |
| 11 | −85,033066 | 10,263433 | Flatland | 93 | 10.08.21 | Limonal |
| 12 | −85,248779 | 10,245504 | Coastal | 18 | 10.08.21 | Near Bridge Guanocaste |
| 13 | −85,535678 | 10,227485 | Flatland | 196 | 10.08.21 | Santa Cruz Vikings |
| 14 | −85,429053 | 10,142110 | Flatland | 112 | 10.08.21 | Nikoyia East road |
| 15 | −85,104708 | 9,960166 | Flatland | 17 | 10.08.21 | After Jikarel |
| 16 | −84,921191 | 9,916594 | Coastal | 5 | 10.08.21 | Playa Blanca Nicoyia |
| 17 | −84,970836 | 9,942300 | Coastal | 8 | 10.08.21 | Playa Naranja Nicoyia |
| 18 | −84,612642 | 9,809249 | Flatland | 67 | 10.08.21 | Cerro Lodge Mirador |
| 19 | −84,715533 | 9,926257 | Coastal | 9 | 10.08.21 | Caldera |
| 20 | −84,614791 | 9,805693 | Flatland | 38 | 11.08.21 | Tarcoles river |
| 21 | −83,909294 | 9,301394 | Flatland | 10 | 11.08.21 | Savegre (near Delta) |
| 22 | −83,733139 | 9,332920 | Mountain | 777 | 11.08.21 | Valle Encantado Restaurant |
| 23 | −83,707093 | 9,385252 | Urban/Peri-Urban | 750 | 11.08.21 | San Isidoro El General |
| 24 | −83,863655 | 9,256796 | Coastal | 10 | 11.08.21 | Rio Dominical |

**Table A1.** *Cont.*

| SITE # | Longitude (WGS84) | Latitude (WGS84) | Landscape Type | Elevation m.a.s.l. | Date | Site Name |
|---|---|---|---|---|---|---|
| 25 | −83,860829 | 9,248910 | Coastal | 2 | 11.08.21 | Playa Dominical |
| 26 | −83,802628 | 9,549204 | Mountain | 2338 | 12.08.21 | Mirador Savegre Hotel |
| 27 | −83,973810 | 9,670681 | Urban/Peri-Urban | 1639 | 13.08.21 | San Rafael Dota |
| 28 | −83,952244 | 9,649735 | Mountain | 1776 | 13.08.21 | Coffee Plantation Dota |
| 29 | −83,957599 | 9,650552 | Mountain | 1686 | 13.08.21 | Don Cayito Dota |
| 30 | −83,977941 | 9,750883 | Mountain | 1945 | 13.08.21 | Cristo Rey Desamparados |
| 31 | −83,838317 | 9,859926 | Urban/Peri-Urban | 1411 | 13.08.21 | Cartago near Birris |
| 32 | −83,562001 | 9,831891 | Mountain | 988 | 14.08.21 | Rancho Naturalista Milking Station |
| 33 | −83,396626 | 10,076883 | Flatland | 81 | 15.08.21 | Sequirres road-Rio Madre de Dios |
| 34 | −82,851667 | 9,624925 | Urban/Peri-Urban | 43 | 15.08.21 | BriBri village |
| 35 | −83,026268 | 9,960614 | Coastal | 5 | 17.08.21 | Limon airport |
| 36 | −83,292599 | 10,045285 | Flatland | 14 | 17.8.21 | Rio Chirippo |
| 37 | −83,798094 | 9,583195 | Mountain | 2602 | 13.08.21 | Dantika |
| 38 | −84,185352 | 9,464873 | Urban/Peri-Urban | 11 | 9.08.21 | Paquita Aguirre (near Parrita) |
| 39 | −82,914773 | 9,787902 | Flatland | 15 | 17.08.21 | Rio Estrella Bonafacio |
| 40 | −83,594367 | 10,156218 | Flatland | 127 | 17.08.21 | Germania |
| 41 | −84,036705 | 10,451382 | Flatland | 65 | 17.08.21 | La Guaria Sarapiqui |
| 42 | −83,507506 | 10,092929 | Urban/Peri-Urban | 90 | 17.08.21 | Siquirres |
| 43 | −82,722680 | 9,644308 | Coastal | 6 | 15.08.21 | Villa Carribe Hotel |
| 44 | −82,755876 | 9,656153 | Urban/Peri-Urban | 4 | 17.08.21 | Stashu's Restaurant PV |
| 45 | −82,649320 | 9,639720 | Coastal | 6 | 16.08.21 | Punta Manzanillio |
| 46 | −84,169478 | 10,345446 | Mountain | 374 | 17.09.21 | Corazon de Jesus |
| 47 | −84,187976 | 10,299510 | Mountain | 753 | 17.09.21 | Laguna Maria Aguilar |
| 48 | −84,192941 | 10,166430 | Mountain | 1977 | 17.08.21 | Paosito Alajuela |
| 49 | −84,251309 | 10,027167 | Urban/Peri-Urban | 892 | 18.08.21 | Villa San Ignacio Alajuela |
| 50 | −84,205267 | 9,997029 | Urban/Peri-Urban | 917 | 18.08.21 | Aeroporto San Juanito Alejuela |

**Table A2.** Site dataset with all scored metrics and LAP conservation index results; site numbers as in Table A1.

| SITE # | Land Use Pattern | Vegetation | Flora | Road Network | Modern Antropogenic Int. | Pollution, Garbage & Debris | Agriculture | Livestock Grazing | Hydorological Alternation | Shorelines &/or Riparian Cond. | Soundscape Quality | Landscape Attractiveness | Smellscape Pleasentness | Wildlife & Wildlife Habitat | Buildings | SUM | Number Filled IN | Index | INDEX Class |
|---|---|---|---|---|---|---|---|---|---|---|---|---|---|---|---|---|---|---|---|
| 1 | 4 | 3 | 3 | 4 | 4 | 4 | | | | 4 | 4 | 7 | 4 | 3 | 5 | 49 | 12 | 41 | POOR |
| 2 | 2 | 2 | 1 | 2 | 3 | 6 | | | | 3 | 1 | 7 | 2 | 2 | 1 | 32 | 12 | 27 | BAD |
| 3 | 8 | 7 | 6 | 8 | 7 | 7 | | | 4 | 5 | 4 | 9 | 2 | 8 | 5 | 80 | 13 | 62 | MODERATE |
| 4 | 9 | 9 | 9 | 9 | 8 | | 8 | | 9 | 9 | 10 | 10 | 10 | 10 | 8 | 118 | 13 | 91 | EXCELLENT |
| 5 | 7 | 7 | 6 | 6 | 5 | 4 | | | | 7 | 3 | 8 | 7 | | 3 | 63 | 11 | 57 | MODERATE |
| 6 | 5 | 6 | 4 | 5 | 5 | | | | 6 | 4 | 3 | 7 | 3 | | 6 | 54 | 11 | 49 | POOR |
| 7 | 4 | 4 | 2 | 5 | 5 | 8 | | | | 4 | 2 | 8 | 3 | 5 | 4 | 54 | 12 | 45 | POOR |
| 8 | 10 | 10 | 10 | 10 | 9 | 10 | | | | 10 | 9 | 10 | 10 | 10 | 8 | 116 | 12 | 97 | EXCELLENT |
| 9 | 3 | 2 | 1 | 4 | 2 | 3 | 1 | 3 | | 2 | 2 | 0 | 1 | 1 | 1 | 25 | 13 | 19 | BAD |
| 10 | 8 | 3 | 2 | 4 | 4 | 4 | 4 | | | | 3 | 1 | 8 | | 6 | 47 | 11 | 43 | POOR |
| 11 | 6 | 3 | 2 | 6 | 6 | | 5 | 3 | | | 6 | 8 | | | | 45 | 9 | 50 | MODERATE |
| 12 | 9 | 9 | 9 | 7 | 5 | 8 | 8 | 6 | 8 | 8 | 4 | 9 | 2 | 10 | 8 | 110 | 15 | 73 | GOOD |
| 13 | 8 | 4 | 4 | | 8 | | 7 | 4 | | | 8 | 7 | | | 4 | 54 | 9 | 60 | MODERATE |
| 14 | 8 | 7 | 7 | 6 | 6 | 9 | 7 | 5 | | | 3 | 5 | | | 5 | 68 | 11 | 62 | MODERATE |
| 15 | 6 | 7 | 6 | 8 | 8 | 9 | | 6 | | | 8 | 9 | 9 | | 9 | 85 | 11 | 77 | GOOD |
| 16 | 9 | 9 | 9 | 8 | 9 | 9 | 10 | | 9 | 8 | 10 | 10 | 10 | 10 | 8 | 128 | 14 | 91 | EXCELLENT |
| 17 | 9 | 9 | 9 | 8 | 7 | 9 | | | | 8 | 5 | 9 | 8 | | 8 | 89 | 11 | 81 | GOOD |
| 18 | 9 | 8 | 7 | 10 | 9 | 10 | 8 | 7 | 10 | 10 | 10 | 10 | 10 | 10 | 9 | 137 | 15 | 91 | EXCELLENT |
| 19 | 4 | 4 | 3 | 4 | 3 | 4 | | | | 4 | 3 | 2 | 7 | 9 | 5 | 52 | 12 | 43 | POOR |
| 20 | 9 | 8 | 7 | 8 | 8 | 10 | 9 | 9 | 9 | 8 | 7 | 10 | 10 | 10 | 8 | 130 | 15 | 87 | EXCELLENT |
| 21 | 8 | 7 | 6 | 8 | 8 | 9 | 5 | | 9 | 3 | | 9 | | | | 72 | 10 | 72 | GOOD |
| 22 | 7 | 7 | 6 | 5 | 7 | 8 | 7 | | 9 | 4 | 6 | 9 | | | 4 | 79 | 12 | 66 | MODERATE |
| 23 | 2 | 2 | 2 | 0 | 2 | 7 | | | | | 1 | 2 | 1 | 2 | 2 | 23 | 11 | 21 | BAD |
| 24 | 4 | 7 | 4 | 4 | 6 | 9 | 6 | | 8 | 3 | 3 | 8 | 8 | 8 | 7 | 85 | 14 | 61 | MODERATE |
| 25 | 8 | 4 | 4 | 5 | 8 | 9 | | | | 8 | 10 | 8 | | | 4 | 68 | 10 | 68 | MODERATE |
| 26 | 8 | 7 | 7 | 7 | 8 | 10 | 8 | 8 | | | 10 | 10 | 10 | 10 | 5 | 108 | 13 | 83 | GOOD |
| 27 | 7 | 4 | 3 | 4 | 4 | 8 | 7 | 8 | | | 4 | 9 | 5 | | 4 | 67 | 12 | 56 | MODERATE |
| 28 | 6 | 6 | 3 | 4 | 8 | 10 | 7 | 8 | | | 7 | 9 | 9 | 3 | 8 | 88 | 13 | 68 | MODERATE |
| 29 | 6 | 3 | 2 | 4 | 6 | 8 | 5 | 6 | | | 4 | 9 | 9 | 2 | 5 | 69 | 13 | 53 | MODERATE |
| 30 | 5 | 5 | 4 | 4 | 4 | 5 | 4 | 4 | | | 3 | 3 | | | 3 | 44 | 11 | 40 | POOR |
| 31 | 2 | 2 | 1 | 3 | 4 | 3 | 2 | 3 | | | 2 | 2 | | 2 | 4 | 30 | 12 | 25 | BAD |
| 32 | 7 | 8 | 6 | 8 | 9 | 10 | 7 | 8 | 5 | | 9 | 10 | 10 | 9 | 7 | 113 | 14 | 81 | GOOD |
| 33 | 6 | 6 | 4 | 5 | 5 | 8 | | | | | 4 | 5 | | | 6 | 49 | 9 | 54 | MODERATE |
| 34 | 8 | 8 | 8 | 6 | 6 | 5 | | | | | 6 | 8 | | 8 | 3 | 66 | 10 | 66 | MODERATE |
| 35 | 4 | 2 | 2 | 3 | 3 | 8 | 2 | | | 3 | 5 | 4 | | 2 | 4 | 42 | 12 | 35 | POOR |
| 36 | 7 | 6 | 6 | 6 | 6 | 9 | 4 | | 8 | 4 | 4 | 7 | | | 9 | 76 | 12 | 63 | MODERATE |
| 37 | 9 | 10 | 7 | 8 | 8 | 10 | 10 | | 10 | 10 | 8 | 10 | 10 | 10 | 7 | 127 | 14 | 91 | EXCELLENT |
| 38 | 2 | 2 | 1 | 3 | 2 | 3 | 1 | | | | 2 | 2 | 2 | 2 | | 22 | 11 | 20 | BAD |
| 39 | 8 | 8 | 8 | 9 | 8 | 9 | 3 | | 9 | 4 | 7 | 8 | | 9 | 10 | 100 | 13 | 77 | GOOD |
| 40 | 5 | 4 | 3 | 2 | 3 | 4 | 1 | | | | 4 | 6 | 2 | 2 | 3 | 39 | 12 | 33 | POOR |
| 41 | 8 | 7 | 6 | 6 | 8 | 9 | 7 | 7 | | | 7 | 7 | 7 | 8 | 5 | 92 | 13 | 71 | GOOD |
| 42 | 4 | 4 | 4 | 3 | 4 | 7 | 2 | | | | 3 | 4 | | | 4 | 39 | 10 | 39 | POOR |
| 43 | 4 | 6 | 6 | 8 | 8 | 9 | 8 | | | 8 | 8 | 10 | 9 | 9 | 6 | 99 | 13 | 76 | GOOD |
| 44 | 5 | 5 | 5 | 6 | 4 | 8 | 9 | | | 5 | 2 | 9 | 3 | 3 | 4 | 68 | 13 | 52 | MODERATE |
| 45 | 10 | 9 | 10 | 10 | 9 | 10 | 10 | | | | 10 | 10 | 10 | 10 | 10 | 118 | 12 | 98 | EXCELLENT |
| 46 | 9 | 8 | 7 | 7 | 7 | 10 | 8 | 6 | | | 10 | 9 | | | 8 | 89 | 11 | 81 | GOOD |
| 47 | 3 | 4 | 2 | 8 | 3 | 9 | | 5 | | 4 | 7 | 4 | | | 5 | 54 | 11 | 49 | POOR |

**Table A2.** *Cont.*

| SITE # | Land Use Pattern | Vegetation | Flora | Road Network | Modern Antropogenic Int. | Pollution, Garbage & Debris | Agriculture | Livestock Grazing | Hydrological Alternation | Shorelines &/or Riparian Cond. | Soundscape Quality | Landscape Attractiveness | Smellscape Pleasentness | Wildlife & Wildlife Habitat | Buildings | SUM | Number Filled IN | Index | INDEX Class |
|---|---|---|---|---|---|---|---|---|---|---|---|---|---|---|---|---|---|---|---|
| 48 | 4 | 2 | 1 | 4 | 6 | 8 | | 2 | | | 7 | 8 | 5 | 2 | 8 | 57 | 12 | 48 | POOR |
| 49 | 2 | 5 | 5 | 1 | 1 | 8 | 2 | | | | 1 | 6 | | 6 | 3 | 40 | 11 | 36 | POOR |
| 50 | 2 | 2 | 1 | 1 | 1 | 9 | | | | | 1 | 4 | 1 | 2 | 4 | 28 | 11 | 25 | BAD |
| **Scored** | 50 | 50 | 50 | 49 | 50 | 46 | 33 | 19 | 15 | 25 | 48 | 50 | 33 | 32 | 47 | | | | |
| **Unscored** | 0 | 0 | 0 | 1 | 0 | 4 | 17 | 31 | 35 | 25 | 2 | 0 | 17 | 18 | 3 | | | | |

**Table A3.** Correlation analyses using Spearman's rho. (Outstanding values are in bold).

| | | Vegetation | Flora | Road Network | Modern Antropogenic Interference | Pollution Garbage & Debris | Agriculture | Livestock Grazing | Hydrological Alteration | Shorelines and/or Riparian Condition | Soundscape Quality | Landscape Attractiveness | Smellscape Pleasantness | Wildlife & Wildlife Habitat | Buildings |
|---|---|---|---|---|---|---|---|---|---|---|---|---|---|---|---|
| **Land Use Pattern** | Correlation Coefficient | 0.824 | 0.835 | 0.776 | 0.777 | 0.578 | 0.738 | 0.590 | 0.665 | 0.773 | 0.700 | 0.682 | 0.710 | 0.825 | 0.595 |
| | Sig.(2-tailed) | 0.000 | 0.000 | 0.000 | 0.000 | 0.000 | 0.000 | 0.008 | 0.007 | 0.000 | 0.000 | 0.000 | 0.000 | 0.000 | 0.000 |
| | N | 50 | 50 | 49 | 50 | 46 | 33 | 19 | 15 | 25 | 48 | 50 | 33 | 32 | 47 |
| **Vegetation** | Correlation Coefficient | | **0.954** | 0.822 | 0.742 | 0.646 | 0.759 | 0.678 | 0.598 | 0.699 | 0.594 | 0.741 | 0.717 | **0.919** | 0.625 |
| | Sig.(2-tailed) | | **0.000** | 0.000 | 0.000 | 0.000 | 0.000 | 0.001 | **0.019** | 0.000 | 0.000 | 0.000 | 0.000 | **0.000** | 0.000 |
| | N | | **50** | 49 | 50 | 46 | 33 | 19 | 15 | 25 | 48 | 50 | 33 | **32** | 47 |
| **Flora** | Correlation Coefficient | | | 0.798 | 0.714 | 0.590 | 0.733 | 0.592 | 0.589 | 0.736 | 0.594 | 0.697 | 0.693 | **0.911** | 0.563 |
| | Sig.(2-tailed) | | | 0.000 | 0.000 | 0.000 | 0.000 | 0.008 | **0.021** | 0.000 | 0.000 | 0.000 | 0.000 | **0.000** | 0.000 |
| | N | | | 49 | 50 | 46 | 33 | 19 | 15 | 25 | 48 | 50 | 33 | **32** | 47 |
| **Road Network** | Correlation Coefficient | | | | 0.812 | 0.655 | 0.749 | 0.442 | 0.572 | 0.701 | 0.748 | 0.744 | 0.742 | 0.839 | 0.696 |
| | Sig.(2-tailed) | | | | 0.000 | 0.000 | 0.000 | **0.066** | **0.026** | 0.000 | 0.000 | 0.000 | 0.000 | 0.000 | 0.000 |
| | N | | | | 49 | 46 | 32 | 18 | 15 | 25 | 47 | 49 | 33 | 32 | 46 |
| **Modern Antropogenic Int.** | Correlation Coefficient | | | | | 0.751 | 0.715 | 0.582 | 0.591 | 0.677 | 0.863 | 0.796 | 0.891 | 0.766 | 0.687 |
| | Sig.(2-tailed) | | | | | 0.000 | 0.000 | 0.009 | **0.020** | 0.000 | 0.000 | 0.000 | 0.000 | 0.000 | 0.000 |
| | N | | | | | 46 | 33 | 19 | 15 | 25 | 48 | 50 | 33 | 32 | 47 |
| **PollutionGarbage & Debris** | Correlation Coefficient | | | | | | 0.728 | 0.729 | 0.602 | 0.559 | 0.707 | 0.708 | 0.801 | 0.689 | 0.687 |
| | Sig.(2-tailed) | | | | | | 0.000 | 0.001 | **0.029** | 0.006 | 0.000 | 0.000 | 0.000 | 0.000 | 0.000 |
| | N | | | | | | 30 | 17 | 13 | 23 | 44 | 46 | 31 | 31 | 44 |
| **Agriculture** | Correlation Coefficient | | | | | | | 0.703 | 0.436 | 0.755 | 0.672 | 0.852 | 0.651 | 0.829 | 0.559 |
| | Sig.(2-tailed) | | | | | | | 0.002 | **0.156** | 0.002 | 0.000 | 0.000 | 0.001 | 0.000 | 0.001 |
| | N | | | | | | | 16 | 12 | 14 | 31 | 33 | 21 | 22 | 30 |

**Table A3.** *Cont.*

| | | Vegetation | Flora | Road Network | Modern Antropogenic Interference | Pollution Garbage & Debris | Agriculture | Livestock Grazing | Hydorological Alteration | Shorelines and/or Riparian Condition | Soundscape Quality | Landscape Attractiveness | Smellscape Pleasantness | Wildlife & Wildlife Habitat | Buildings |
|---|---|---|---|---|---|---|---|---|---|---|---|---|---|---|---|
| | | | | | | | | | | | | | | | |
| | | | | | | | | | **Spearman's Rho Correlations** | | | | | | |
| LivestockGrazing | Correlation Coefficient Sig.(2-tailed) N | | | | | | | | 0.000 **1.000** 4 | 0.632 **0.368** 4 | 0.485 **0.041** 18 | 0.754 0.000 19 | 0.636 **0.020** 13 | 0.687 **0.019** 11 | 0.397 **0.103** 18 |
| HydorologicalAlternation | Correlation Coefficient Sig.(2-tailed) N | | | | | | | | | 0.581 **0.029** 14 | 0.687 0.007 14 | 0.563 **0.029** 15 | 0.667 **0.035** 10 | 0.715 **0.013** 11 | 0.492 **0.074** 14 |
| ShorelinesamporRiparianCond | Correlation Coefficient Sig.(2-tailed) N | | | | | | | | | | 0.709 0.000 24 | 0.784 0.000 25 | 0.727 0.001 17 | 0.815 0.000 17 | 0.456 **0.025** 24 |
| SoundscapeQuality | Correlation Coefficient Sig.(2-tailed) N | | | | | | | | | | | 0.702 0.000 48 | 0.866 0.000 32 | 0.727 0.000 32 | 0.608 0.000 46 |
| LandscapeAttractiveness | Correlation Coefficient Sig.(2-tailed) N | | | | | | | | | | | | 0.834 0.000 33 | 0.828 0.000 32 | 0.589 0.000 47 |
| SmellscapePleasentness | Correlation Coefficient Sig.(2-tailed) N | | | | | | | | | | | | | 0.790 0.000 26 | 0.702 0.000 32 |
| Wildlife&WildlifeHabitat | Correlation Coefficient Sig.(2-tailed) N | | | | | | | | | | | | | | 0.714 0.000 31 |

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
