# Peer review of "Landscape Conservation Assessment in the Latin American Tropics: Application and Insights from Costa Rica"

_land, doi:10.3390/land11040514_

Round 1

Reviewer 1 Report

Dear Authors,

Thank you for answering my questions. However, there are a few more points that I believe need revision.

  1. In my opinion, the description of Costa Rica's environment is still insufficient. There is no basic information in the manuscript on the topography and land cover (plant communities). These two elements of the environment are crucial in terms of landscape diversity. Additionally, not all readers of the manuscript are familiar with enovironemnt of Costa Rica. In my opinion, such basic information should be presented.
  2. Your answer to the question related to the impact of range of view on the assessment was: "Response: Generally most landscapes that were assessed where high relief landscapes with much vertical range. This is not a place to compare."However, the analysis of figure 3 shows that about half of the assessed landscapes are "coastal" and "flatland". The difference in relative heights and the range of visibility is smaller in such landscapes. The question I posed in the first review is therefore still valid.
  3. I was asking about the assessment of coastal landscapes. And there is no response to my question. Figure 8b shows a landscape whose main element is the sea surface. In this case, how can the criteria be assessed: lands use pattern, road network etc.? Please explain whether coastal landscapes with large water surfaces (sea) were assessed.

Author Response

1

 Open Review

English language and style

( ) Extensive editing of English language and style required
( ) Moderate English changes required
(x) English language and style are fine/minor spell check required
( ) I don't feel qualified to judge about the English language and style

Yes

Can be improved

Must be improved

Not applicable

Does the introduction provide sufficient background and include all relevant references?

(x)

( )

( )

( )

Is the research design appropriate?

(x)

( )

( )

( )

Are the methods adequately described?

(x)

( )

( )

( )

Are the results clearly presented?

(x)

( )

( )

( )

Are the conclusions supported by the results?

(x)

( )

( )

( )

Comments and Suggestions for Authors

Dear Authors,

Thank you for answering my questions. However, there are a few more points that I believe need revision.

  1. In my opinion, the description of Costa Rica's environment is still insufficient. There is no basic information in the manuscript on the topography and land cover (plant communities). These two elements of the environment are crucial in terms of landscape diversity. Additionally, not all readers of the manuscript are familiar with enovironemnt of Costa Rica. In my opinion, such basic information should be presented.

AUTHORS’ RESPONSE: The new paragraph that was added in the revised manuscript has been augmented and there are a critically important references that provide all that the reviewer suggests should be given (i.e. Kappelle, M. (2016). Costa Rican Ecosystems. Chicago, Illinois. University of Chicago Press). New sentences on topography and vegetation have been added. Also, note that figure 8 which showcases the said ecosystems and landscapes providing the reader with images of the vegetation etc.

  1. Your answer to the question related to the impact of range of view on the assessment was: "Response: Generally most landscapes that were assessed where high relief landscapes with much vertical range. This is not a place to compare."However, the analysis of figure 3 shows that about half of the assessed landscapes are "coastal" and "flatland". The difference in relative heights and the range of visibility is smaller in such landscapes. The question I posed in the first review is therefore still valid.

AUTHORS’ RESPONSE:  Firstly, there are high relief conditions in many coastal areas particularly along the west coast of CR; so many of the coastal areas do have varied topography and a curtain of mountains as a backdrop. Flatland areas are areas where flat land- plateaus and plains dominate, but again the background mountains are often visible. IN this sense much of the landscape is definitely “high relief” and heterogeneous as compared to countries/regions where plains dominate. This is what we meant in our response.

The issue of accessibility from which the assessments were made was not the only criterion in selecting places to assess landscape views, but it was impossible to do better within a search-find expedition in some parts of the country. We also wanted to find impacted areas, we are not attempting to organize a complete survey, we are attempting to have representative landscape areas as best we could. We feel we have a fairly typical set of 50 landscape views.

We thank this reviewer for focusing on this difficulty and have made the following addition in the text on this issue:

The section “2.3. Application in Costa Rica: specific on-site methods” has been split into two paragraphs and additions have been made as suggested by this reviewer.

  1. I was asking about the assessment of coastal landscapes. And there is no response to my question. Figure 8b shows a landscape whose main element is the sea surface. In this case, how can the criteria be assessed: lands use pattern, road network etc.? Please explain whether coastal landscapes with large water surfaces (sea) were assessed.

AUTHORS’ RESPONSE:  We are sorry not to reply; in our minds this is obvious: areas of the water surface itself are natural features (marine sea areas)-subsets of the landscape or backdrops of the landscape; these features cannot be assessed as terrestrial landscapes.

Other than early work on landscape conservation designations, the idea of seascapes is not given the same weight of landscapes in the conservation science literature. Seascape is used to include areas where the “water feature” may dominate, but it often includes island clusters, archipelagos, and terrestrial coastal zones. In a landscape sense deal with the marine water surface as a natural feature only.

Please see: Phillips, Adrian, (2002). Management Guidelines for IUCN Category V Protected Areas: Protected Landscapes/Seascapes. IUCN Gland, Switzerland and Cambridge, UK. xv + 122pp.

When applying LAP (as in the Figure 8b) the land-use pattern, road network etc metrics obviously apply only where they can be perceived by the assessor, i.e. on land.

Submission Date

01 February 2022

Date of this review

07 Feb 2022 08:45:57

Reviewer 2 Report

Dear authors, you are presenting a very interesting paper. I indicated my corrections in the attached PDF, please add all corrections in order to improve your manuscript.

All the best.

Author Response

2

Open Review

English language and style

( ) Extensive editing of English language and style required
( ) Moderate English changes required
(x) English language and style are fine/minor spell check required
( ) I don't feel qualified to judge about the English language and style

AUTHORS’ RESPONSE: Extensive english corrections have been done throughout text of this revised MS.

Yes

Can be improved

Must be improved

Not applicable

Does the introduction provide sufficient background and include all relevant references?

( )

( )

(x)

( )

Is the research design appropriate?

(x)

( )

( )

( )

Are the methods adequately described?

(x)

( )

( )

( )

Are the results clearly presented?

( )

( )

(x)

( )

Are the conclusions supported by the results?

(x)

( )

( )

( )

Comments and Suggestions for Authors

Dear authors, you are presenting a very interesting paper. I indicated my corrections in the attached PDF, please add all corrections in order to improve your manuscript.

All the best.

AUTHORS’ RESPONSE:  Nearly all corrections suggested have been incorporated and we include notes/replies to the original marked pdf manuscript. We have added most new suggested references. We thank this reviewer for all work done to assist in amelioration of our paper and agree with nearly all his/hers very valuable suggestions.

Submission Date

01 February 2022

Date of this review

14 Feb 2022 17:25:24

3

Open Review

English language and style

(x) Extensive editing of English language and style required
( ) Moderate English changes required
( ) English language and style are fine/minor spell check required
( ) I don't feel qualified to judge about the English language and style

AUTHORS’ RESPONSE: The entire MS was re-read by a proficient native English speaker and ameliorations were made throughout (evident in the tracked-changes revised document). We tried to make the work more concise and have cleared out jargon – to ameliorate for clarity.

Yes

Can be improved

Must be improved

Not applicable

Does the introduction provide sufficient background and include all relevant references?

( )

( )

(x)

( )

Is the research design appropriate?

( )

(x)

( )

( )

Are the methods adequately described?

( )

( )

(x)

( )

Are the results clearly presented?

( )

( )

(x)

( )

Are the conclusions supported by the results?

( )

( )

(x)

( )

Comments and Suggestions for Authors

When mentioning Latin America, the title does not reflect the accurate spatial scale of the research, so I suggest limiting the title to Costa Rica.

AUTHORS’ RESPONSE: This is a question that has seriously involved much thought. Firstly, we did change the title from the original manuscript in the previously revised version (i.e. from “Latin America” to “tropical Latin America”). We are applying a survey method and index in the humid tropics within the neotropical biogeographical realm, we are not focusing on its used within one particular geographic region or state. This is why we feel the current title is appropriate. Thanks for mentioning this; and this is the only reviewer who had concern for this.

The research question does not exist in the introduction, nor does the hypothesis. The absence of the research question or questions does not allow evaluating whether or not the results answer the research question.

AUTHORS’ RESPONSE: We have re-worded our original last paragraph of the introduction to help make the research question more specific. This is now presented as such:

We decided to explore landscape quality in a tropical humid climate area through an on-site approach using the Landscape Assessment Protocol (LAP), a rapid landscape assessment survey method [57 Vlami et al. 2019]. This method has been developed and tested primarily in temperate and Mediterranean climate areas and was first published in 2016 [58 Vlami et al., 2016] and has not been formally tested in the tropics. Here, we aimed to apply a country-wide survey of landscape views using the original LAP method and we test and critique the application in a wide variety of landscape types in Costa Rica. We aim to see if there are benefits in utilizing such a rapid assessment tool for baseline inventory of landscape conditions, landscape degradation description and how this may be of use in landscape conservation in the Latin American tropics.

Methodology:
The number of interviews could be very few for a national scale and the assortment of landscapes of Costa Rica. In addition, it added to the high biodiversity, cultures, species, and ecosystems. Therefore, it is necessary to include an analysis that statistically justifies the number of interviews.

AUTHORS’ RESPONSE: No interviews were made; our work is based on a rapid assessment protocol method; please read Vlami et al. 2019 for the general philosophy and rationale behind this approach.

It is necessary to explain the selection bias the results could have when considering a single informant or interviewee per landscape or survey.
Explain how they will block preference among respondents related to gender, age, education level, race. For example, a change or bias in these variables could lead to different research results.

I consider that descriptive statistics do not contribute much to a robust statistical analysis. However, it is necessary to include more statistical analysis that addresses qualitative methods.

AUTHORS’ RESPONSE: Again, no interviews were made. Our work provides a large number of statistical approaches to describe and interpret such a dataset of on-site landscape view assessments.

The discussion is a mix of a description of methods and a literature review. It is necessary to synthesize and discuss the research results with other studies. For instance, line 502 - 514

It is necessary to include one section on the method's limitations.

AUTHORS’ RESPONSE: WE have carefully restructured large parts of the discussion including LINES 504-514 in order to make the discussion approach clearer.

Limitations of the method are carefully and specifically developed within the discussion:

The conclusion is not robust addresses more recommendations or limitations.

AUTHORS’ RESPONSE: We have carefully restructured the conclusion and hope this satisfies; we do not want to repeat aspects that are developed in the discussion (where there area specific developments on recommendations for future actions and the method’s limitations and difficulties (i.e subjectivity etc.).

Submission Date

01 February 2022

Date of this review

18 Feb 2022 15:20:25

Reviewer 3 Report

When mentioning Latin America, the title does not reflect the accurate spatial scale of the research, so I suggest limiting the title to Costa Rica.

The research question does not exist in the introduction, nor does the hypothesis. The absence of the research question or questions does not allow evaluating whether or not the results answer the research question.

Methodology:
The number of interviews could be very few for a national scale and the assortment of landscapes of Costa Rica. In addition, it added to the high biodiversity, cultures, species, and ecosystems. Therefore, it is necessary to include an analysis that statistically justifies the number of interviews.

It is necessary to explain the selection bias the results could have when considering a single informant or interviewee per landscape or survey.
Explain how they will block preference among respondents related to gender, age, education level, race. For example, a change or bias in these variables could lead to different research results.

I consider that descriptive statistics do not contribute much to a robust statistical analysis. However, it is necessary to include more statistical analysis that addresses qualitative methods.

The discussion is a mix of a description of methods and a literature review. It is necessary to synthesize and discuss the research results with other studies. For instance, line 502 - 514

It is necessary to include one section on the method's limitations.

The conclusion is not robust addresses more recommendations or limitations.

Author Response

3

Open Review

English language and style

(x) Extensive editing of English language and style required
( ) Moderate English changes required
( ) English language and style are fine/minor spell check required
( ) I don't feel qualified to judge about the English language and style

AUTHORS’ RESPONSE: The entire MS was re-read by a proficient native english speaker and ameliorations were made throughout (evident in the tracked-changes revised document). We tried to make the work more concise and have cleared-out lots of jargon – to ameliorate for clarity and to make the work more concise.

Yes

Can be improved

Must be improved

Not applicable

Does the introduction provide sufficient background and include all relevant references?

( )

( )

(x)

( )

Is the research design appropriate?

( )

(x)

( )

( )

Are the methods adequately described?

( )

( )

(x)

( )

Are the results clearly presented?

( )

( )

(x)

( )

Are the conclusions supported by the results?

( )

( )

(x)

( )

Comments and Suggestions for Authors

When mentioning Latin America, the title does not reflect the accurate spatial scale of the research, so I suggest limiting the title to Costa Rica.

AUTHORS’ RESPONSE: This is a question that has seriously involved much thought. Firstly, we did change the title from the original manuscript in the previously revised version (i.e. from “Latin America” to “tropical Latin America”). We are applying a survey method and index in the humid tropics within the neotropical biogeographical realm, we are not focusing on its used within one particular geographic region or state. This is why we feel the current title is appropriate. Thanks for mentioning this; and this is the only reviewer who had concern for this.

The research question does not exist in the introduction, nor does the hypothesis. The absence of the research question or questions does not allow evaluating whether or not the results answer the research question.

AUTHORS’ RESPONSE: We have re-worded our original last paragraph of the introduction to help make the research question more specific. This is now presented as such:

"We decided to explore landscape quality in a tropical humid climate area through an on-site approach using the Landscape Assessment Protocol (LAP), a rapid landscape assessment survey method [57 Vlami et al. 2019]. This method has been developed and tested primarily in temperate and Mediterranean climate areas and was first published in 2016 [58 Vlami et al., 2016] and has not been formally tested in the tropics. Here, we aimed to apply a country-wide survey of landscape views using the original LAP method and we test and critique the application in a wide variety of landscape types in Costa Rica. We aim to see if there are benefits in utilizing such a rapid assessment tool for baseline inventory of landscape conditions, landscape degradation description and how this may be of use in landscape conservation in the Latin American tropics".

Methodology:
The number of interviews could be very few for a national scale and the assortment of landscapes of Costa Rica. In addition, it added to the high biodiversity, cultures, species, and ecosystems. Therefore, it is necessary to include an analysis that statistically justifies the number of interviews.

AUTHORS’ RESPONSE: No interviews were made; our work is based on a rapid assessment protocol method; please read Vlami et al. 2019 for the general philosophy and rationale behind this approach.

It is necessary to explain the selection bias the results could have when considering a single informant or interviewee per landscape or survey.
Explain how they will block preference among respondents related to gender, age, education level, race. For example, a change or bias in these variables could lead to different research results.

I consider that descriptive statistics do not contribute much to a robust statistical analysis. However, it is necessary to include more statistical analysis that addresses qualitative methods.

AUTHORS’ RESPONSE: Again, no interviews were made. Our work provides a large number of statistical approaches to describe and interpret such a dataset of on-site landscape view assessments.

The discussion is a mix of a description of methods and a literature review. It is necessary to synthesize and discuss the research results with other studies. For instance, line 502 - 514

It is necessary to include one section on the method's limitations.

AUTHORS’ RESPONSE: We have carefully restructured large parts of the discussion including LINES 504-514 in order to make the discussion approach clearer. Limitations of the method are carefully and specifically developed within the discussion. Section 4.2 gives specific details of the potential weaknesses (subjectivity of some metrics) and difficulty to percieve aspects of the landscape attributes (this is also given in section 4.1). 

The conclusion is not robust addresses more recommendations or limitations.

AUTHORS’ RESPONSE: We have carefully restructured the conclusion and hope this satisfies; we do not want to repeat aspects that are developed in the discussion (where there area specific developments on recommendations for future actions and the method’s limitations and difficulties (i.e subjectivity etc.).

Submission Date

01 February 2022

Date of this review

18 Feb 2022 15:20:25

Round 2

Reviewer 3 Report

I think that the answers of the authors satisfy my comments.

Best,

This manuscript is a resubmission of an earlier submission. The following is a list of the peer review reports and author responses from that submission.

Round 1

Reviewer 1 Report

This article discusses a Landscape Assessment Protocol performed in Costa Rica as a rapid-inventory methodology for understanding conservation priority areas within working landscapes. The topic has potential and the researchers seem to understand the value of the research better than they communicate. However, there are some fundamental flaws that need to be worked out before it can be considered for publication. I outline these below and I hope you will take them in a positive spirit, as I believe there is the potential for an interesting contribution in your data but at present in needs quite a bit of reworking.

  1. You take the approach of “throw everything at the wall and see what sticks.” The problem is, everything is a bit underdeveloped and it isn’t clear how this research contributes to the literature in the proposed fields: landscape ecology, conservation science, ecosystem services assessment. I suspect that your strongest thread is the one that you lean on the most – that a LAP is akin to the SVAP and as such offers a novel, relatively accessible, rapid methodology for understanding the conservation value of landscapes (and possibly subsequent intervention). If you decide to go this route – you really need to double down on the value of the methodology, explain the methodology adequately (see note below), and explain how it tells us things we can’t learn from alternative methods (on-ground rapid-assessments, remote-sensing, mixed-methods). It seems that you have developed the value of the methodology elsewhere, applied it to the neotropics, but you have not yet made a convincing case for why it is a useful new contribution.
  2. You also talk about the educational value, citizen science, multifunctional landscapes, policy intervention, and continually repeat an erroneous idea (that perhaps you are just communicating insufficiently as I don’t doubt that you know the topic better than you express) that “little is known about Costa Rican landscapes” and that “little attention has been paid to them.” This is simply not true. Costa Rica is probably the best studied country in the neotropics, with the best studied conservation efforts and policies and your literature review on these specifics is very slim. Furthermore, the relatively recent shift in Costa Rica to focus on biological corridors explicitly emphasizes these mixed-use landscapes, and there is a burgeoning literature focusing on these units of analysis. You cite a lot of older foundational conservation work (Boza, Janzen, Harvey), and your own work, but there is a wealth of literature on efforts to promote sustainable working landscapes in Costa that you would need to include to make some of the claims for how your research amplifies or improves what we already know. The same goes for your discussion of policies. Be cautious with statements that claim that Costa Rica is a country where the landscape is “not yet given policy-relevance for resource protection.” Though the biological corridors initiative lacks teeth, that is precisely what it is. Some of the policies you seem aware of but don’t address (ecotourism financing and infrastructure support) and others you don’t reference (PES, ley de caza, ley forestal independent of PES, institutional restructuring within SINAC and MINAE). This thread is much weaker in your paper and I don’t actually suggest you attempt to do it justice, but you would then need to remove all of the side discussions to this effect, unless you are going to dive into some of the work – it all depends on how you work out #2 above.
  3. The methods are not adequately described. I do not fully understand what you did. It is not clear how you chose sites (though you do discuss some of the challenges) and whether these sites might at all be representative of the phenomena that you hope to understand. I do not understand how the matrix was generated. The methods do not describe the statistical analyses (you have some of this in your results, but it should be moved to the methods with a brief explanation / defense of your approach to what you evaluated and why). The most insight occurred with the reference to the SVAP, as I am familiar with that protocol and could imagine the application to the landscape and somewhat follow along with the ranking system. If your contribution is methodological, you need to be very explicit in describing methods, and show in your results how this increases our understanding of landscapes.
  4. The results are very difficult to follow. You show the reader all your results, but don’t tell us what is important and why. There needs to be more text that walks the reader through the elements that are important so that we know what to focus on. This is particularly problematic given 1-3 above. I also don’t understand why figure 8 is in the discussion and not the results. Note that the discussion should emphasize the implications of the key components of your results. Because there are not key components pointed out (I can identify a few interesting trends but it is all buried in the narrative), the multiple topics in the discussion are again particularly problematic.

The last note is that you reference it being a “pilot study” quite a few times. This is fine, as often pilot studies generate enough quality data to be publishable. However, you use that term as a defense for not having done some of the things with your methods that seem to need to be done to make a clear contribution – especially because the most novel element of your study is the application of the LAP to the neotropics, its validity, and what we can learn. I suggest you reassess and really sell this article as a methodological contribution - and highlight why this builds on the other (numerous) ways that researchers have tried to understand working landscapes in the region.

Reviewer 2 Report

Dear Authors,

I read the manuscript with interest. A comprehensive assessment of the quality / conservation of the landscape is a big challenge. This problem is an important research topic in landscape ecology. Below are some of my comments that may improve the quality of the manuscript.

  1. It seems to me that the introduction should pay more attention to state-of-the-art in the field of expert assessments of landscape quality. The method proposed in the manuscript is one of many and it seems that it should be clearly indicated.
  2. The description of the methods needs to be supplemented. Please describe the assessment criteria and the scoring rules. The reader does not know what the criteria "Vegetation", "Flora" or "Road network" mean. When a given landscape is rated "1" and when "5" in a given criterion. Without such information, it is difficult to assess the degree of subjectivity of assessment, which is crucial for the obtained results.
  3. It seems to me that the manuscript should contain a short description of the environment of Costa Rica, mainly topography, land cover, and anthropogenic elements that influence the development of specific landscapes. 
  4. The characteristics of the geographical environment, especially topography, would allow the assessment of whether the selected landscapes are representative. Costa Rica is an upland-mountain country, therefore I have some doubts whether the coastal and lowland landscapes are not overrepresented in the research. The issue of accessibility from which the assessments were made cannot be the only criterion.
  5. I would like to ask you about the difference in the vertical range of visibility in individual landscapes. How does it affect the assessment results?
  6. The assessment of coastal landscapes may be somewhat questionable, especially as the principles of the assessment are not clearly described. Have landscapes with a significant share of the sea surface been assessed? Or does the term coastal landscape refer to places located in the coastal zone but at a certain distance from the coastline?
  7. The averaged results for landscape quality are quite obvious. Mountain and coastal landscapes received the highest score, and anthropogenic landscapes received the lowest. What is the scientific and practical value of the obtained results?
  8. In my opinion, the Discussion should refer to the problem of transferring the assessment of landscape quality at the point provided by the applied method to larger areas (spatial dimension of the assessment). This would make it possible to use its results for landscape management.